# Emotional Face-to-Speech

Jiaxin Ye [1]   Boyuan Cao [1]   Hongming Shan [1]

## Abstract

How much can we infer about an emotional voice solely from an expressive face? This intriguing question holds great potential for applications such as virtual character dubbing and aiding individuals with expressive language disorders. Existing face-to-speech methods offer great promise in capturing identity characteristics but struggle to generate diverse vocal styles with emotional expression. In this paper, we explore a new task, termed *emotional face-to-speech*, aiming to synthesize emotional speech directly from expressive facial cues. To that end, we introduce **DEmoFace**, a novel generative framework that leverages a discrete diffusion transformer (DiT) with curriculum learning, built upon a multi-level neural audio codec. Specifically, we propose multimodal DiT blocks to dynamically align text and speech while tailoring vocal styles based on facial emotion and identity. To enhance training efficiency and generation quality, we further introduce a coarse-to-fine curriculum learning algorithm for multi-level token processing. In addition, we develop an enhanced predictor-free guidance to handle diverse conditioning scenarios, enabling multi-conditional generation and disentangling complex attributes effectively. Extensive experimental results demonstrate that DEmoFace generates more natural and consistent speech compared to baselines, even surpassing speech-driven methods. Demos of DEmoFace are shown at our project https://demoface.github.io.

[1]Institute of Science and Technology for Brain-Inspired Intelligence, MOE Frontiers Center for Brain Science, Key Laboratory of Computational Neuroscience and Brain-Inspired Intelligence, and State Key Laboratory of Brain Function and Disorders, Fudan University, Shanghai, China. Correspondence to: Hongming Shan <hmshan@fudan.edu.cn>.

*Proceedings of the 42nd International Conference on Machine Learning*, Vancouver, Canada. PMLR 267, 2025. Copyright 2025 by the author(s).

## 1. Introduction

When we encounter a person's face on platforms like Instagram or Facebook without hearing their voice, our minds instinctively generate auditory expectations based on visual cues. These expectations are shaped by our experiences and cultures, influencing how we perceive individuals to sound based on their external appearance, such as age, gender, nationality, or emotion (Paris et al., 2017; Taitelbaum-Swead & Fostick, 2016). These preconceived notions drive us to form judgments about others' voices even before they speak.

In recent years, face-guided Text-to-Speech (TTS) (Goto et al., 2020; Kang et al., 2023; Jang et al., 2024), also known as Face-to-Speech or F2S, has attracted growing interest with diverse applications, such as virtual character dubbing and assistance for individuals with expressive language disorders. The goal of the F2S is to create voices that are consistent with the guided face. However, users increasingly expect generated speech that not only replicates speakers' identities but also conveys rich emotional prosody, enhancing their experience in human-machine interactions (Wen et al., 2024; Cong et al., 2024a; Ye et al., 2023b). This expectation is beyond the scope of existing F2S tasks, lacking explicit guidance to produce desired emotional speech.

Considering that facial expressions are the most direct indicators of emotion, we propose an extension to the F2S task grounded in visual cues, termed **emotional Face-to-Speech (eF2S)**. An example of the proposed eF2S task is illustrated in Fig. 1(b). Unlike the conventional F2S task, which converts text to speech guided solely by identity embeddings extracted from a reference face, our eF2S task further decouples identity and emotion from the facial input, producing speech that preserves the speaker's identity while enriching it with the emotional expression derived from the reference face. While the new eF2S task may initially appear to only require the generated speech to convey emotions, it raises several novel challenges. On the task level, traditional F2S tasks are insufficient for eF2S, as they focus on converting text into speech that reflects facial characteristics without considering emotional states. On the method level, previous expressive speech generation methods focus on generating speech tied to either a specific identity or a specific emotion, but customizing both simultaneously remains a significant challenge—particularly in the absence of speech cues.

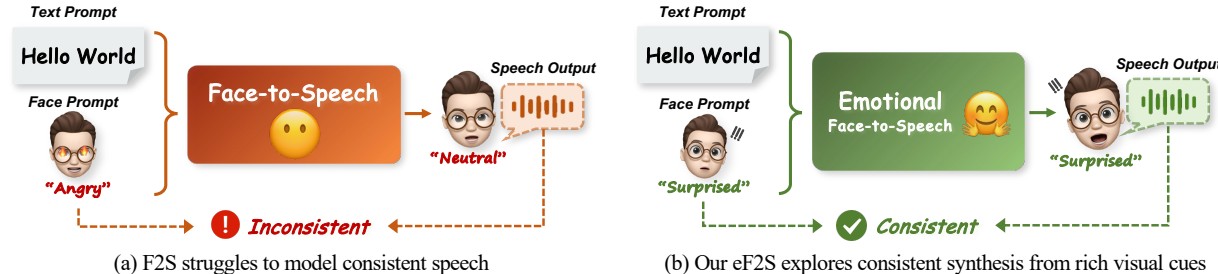

(a) F2S struggles to model consistent speech    (b) Our eF2S explores consistent synthesis from rich visual cues

Figure 1: **Tasks comparison.** (a) Conventional Face-to-Speech (F2S). (b) The introduced Emotional Face-to-Speech (eF2S). Given text and face prompts, the model is expected to generate speech that aligns with both the facial identity and emotional expression. Our eF2S offers a novel perspective for generating consistent speech without relying on any vocal cues.

To address these challenges, we propose a novel discrete **D**iffusion framework for **Emo**tional **Face**-to-speech, called **DEmoFace**, which is the first attempt to generate consistent auditory expectations (*i.e.* identity and emotions) directly from visual cues. Specifically, to mitigate the one-to-many issues inherent in continuous speech features (Xue et al., 2023), we begin by discretizing the speech generation process utilizing neural audio codec with Residual Vector Quantization (RVQ). Considering the low-to-high frequency distributions across different RVQ levels, we then introduce a discrete diffusion model with a coarse-to-fine curriculum learning strategy, to enhance both training efficiency and diversity of generated speech. Building on this, we propose a multimodal DiT (MM-DiT) for the reverse diffusion process, which dynamically aligns speech and text prompts, and customizes face-style linguistic expressions. Furthermore, for challenging multi-conditional generation of eF2S, we develop enhanced predictor-free guidance (EPFG) on the discrete diffusion model, boosting efficient response to the global condition while facilitating the decoupling of multiple local conditions. Comprehensive experiments demonstrate that DEmoFace achieves diverse and high-quality speech synthesis, outperforming the state-of-the-art models in naturalness and consistency— exceeding even speech-guided methods. The contributions of this paper are summarized below:

- We introduce an extension to F2S task, named Emotional Face-to-Speech (eF2S), which is the first attempt to customize a consistent vocal style solely from the face.

- We propose a novel discrete diffusion framework for speech generation, which incorporates multimodal DiT and RVQ-based curriculum learning, achieving high-fidelity generation and efficient training.

- We devise enhanced predictor-free guidance to boost sampling quality in multi-conditional scenarios of eF2S.

- Extensive experimental results demonstrate that DEmoFace can generate more consistent, natural speech with enhanced emotions compared to previous methods.

## 2. Related Work

### 2.1. Residual Vector Quantization for TTS

Neural audio codecs (Zeghidour et al., 2022; Zhang et al., 2024) enable discrete speech modeling by reconstructing high-quality audio at low bitrates. Residual Vector Quantization (RVQ) (Vasuki & Vanathi, 2006; Zheng et al., 2024) is a standard technique for codecs that quantizes audio frames by multiple hierarchical layers of quantizers, with each subsequent layer working on the residual of the previous stage. Recent models (Kharitonov et al., 2023; Shen et al., 2024) rely on codecs with the RVQ to synthesize speech, and show promising performance on naturalness. For example, VALL-E (Wang et al., 2023) employs Encodec (Défossez et al., 2023) to transform the speech into a sequence of discrete tokens, then uses an auto-regressive (AR) model to predict tokens based on phonemes and speaker prompts. However, existing methods are mainly based on the AR manner, leading to unstable and inefficient sampling. We propose a discrete diffusion framework DEmoFace to reconstruct tokens from the RVQ codec, achieving faster and more diverse sampling by parallel iterative refinement.

### 2.2. Face-driven TTS

Face-driven TTS (F2S) aims to synthesize speech based on visual information about the speaker (Plüster et al., 2021; Lee et al., 2024; Goto et al., 2020; Kang et al., 2023; Ye & Shan, 2025). Previous F2S methods focus on how to learn visual representation from speech supervision. For example, Goto *et al.* (Goto et al., 2020) propose a supervised generalized end-to-end loss to minimize the distance between visual embedding and vocal speaker embedding. However, existing methods ignore the rich emotional cues inherent in the face, which often generate over-smoothing speech lacking diverse emotional naturalness. Although Kang *et al.* (Kang et al., 2023) additionally introduce speech prosody codes to enhance the naturalness. They still depend on the speech prompt to achieve natural generation, which does not satisfy the requirements of eF2S. In contrast, our DEmo-

Face only utilizes visual cues to form the emotional auditory expectations without relying on any vocal features.

## 2.3. Emotional TTS

Emotional TTS aims to enhance synthesized speech with emotional expressiveness (Li et al., 2024; Guo et al., 2023b; Chen et al., 2022). Existing methods can be divided into two categories based on how to integrate emotion information into TTS systems. For emotion label conditioning (Ye et al., 2023a; Wen et al., 2022; Ye et al., 2022), EmoDiff (Guo et al., 2023b) introduces a diffusion model with soft emotion labels as a classifier guidance. In contrast, V2C-Net (Chen et al., 2022) employs emotion and speaker embeddings from reference face and speech individually for speech customization. However, previous methods do not explore how to learn both speaker identity and emotion from the face image. We offer a novel perspective of the relationship between auditory expectations and visual cues for TTS without relying on any vocal cues.

# 3. Preliminary: Discrete Diffusion Models

Continuous Diffusion Models (CDM) (Blattmann et al., 2023; Li et al., 2023; Ruan et al., 2023) have achieved state-of-the-art results in generative modeling, but face challenges in speech generation due to high-dimensional speech features and excessive diffusion steps, frustrating practical application. The fundamental solution lies in compressing the speech feature space, such as a discrete space.

Recently, Discrete Diffusion Models (DDMs) have shown promise in language modeling (Meng et al., 2022; Lou et al., 2024) and speech generation (Yang et al., 2023; Wu et al., 2024). We emphasize that DDM has yet to be explored in multi-conditional speech generation with high-quality audio compression. In this paper, to our knowledge, we take the first attempt to generate RVQ-based speech tokens with DDM. Below, we outline the forward and reverse processes of the DDM, along with its training objective.

**Forward diffusion process.** Given a sequence of tokens $\boldsymbol{x} = x^1 \ldots x^d$ from a state space of length $d$ like $\mathcal{X}^d = \{1, \ldots, n\}^d$. The continuous-time discrete Markov chain at time $t$ is characterized by the diffusion matrix $\boldsymbol{Q}_t \in \mathbb{R}^{n^d \times n^d}$ (*i.e.* transition rate matrix), as follows:

$$p(x_{t+\Delta t}^i | x_t^i) = \delta_{x_{t+\Delta t}^i x_t^i} + \boldsymbol{Q}_t(x_{t+\Delta t}^i, x_t^i)\Delta t + o(\Delta t), \quad (1)$$

where $x_t^i$ denotes $i$-th element of $\boldsymbol{x}_t$, $\boldsymbol{Q}_t(x_{t+\Delta t}^i, x_t^i)$ is the $(x_{t+\Delta t}^i, x_t^i)$ element of $\boldsymbol{Q}_t$, denoting the transition rate from state $x_t^i$ to state $x_{t+\Delta t}^i$ at time $t$, and $\delta$ is Kronecker delta. Since the exponential size of $\boldsymbol{Q}_t$, existing works (Lou et al., 2024; Ou et al., 2024) propose to assume dimensional independence, conducting a one-dimensional diffusion process

for each dimension with the same token-level diffusion matrix $\boldsymbol{Q}_t^{\text{tok}} = \sigma(t)\boldsymbol{Q}^{\text{tok}} \in \mathbb{R}^{n \times n}$, where $\sigma(t)$ is the noise schedule and $\boldsymbol{Q}^{\text{tok}}$ is designed to diffuse towards an absorbing state [MASK]. Then the forward equation is formulated as $\boldsymbol{P}(x_t^i, x_0^i) = \exp\left(\bar{\sigma}(t)\boldsymbol{Q}^{\text{tok}}(x_t^i, x_0^i)\right)$, where transition probability matrix $\boldsymbol{P}(x_t^i, x_0^i) := p(x_t^i | x_0)$, and cumulative noise $\bar{\sigma}(t) = \int_0^t \sigma(s)ds$. There are two probabilities in the $\boldsymbol{P}_{t|0}$: $1 - e^{-\bar{\sigma}(t)}$ for replacing the current tokens with [MASK], $e^{-\bar{\sigma}(t)}$ for keeping it unchanged.

**Reverse denoising process.** As the diffusion matrix $\boldsymbol{Q}_t^{\text{tok}}$ is known, the reverse process can be given by a reverse transition rate matrix $\bar{\boldsymbol{Q}}_t$ (Sun et al., 2023; Kelly, 2011), where $\bar{\boldsymbol{Q}}_t(x_{t-\Delta t}^i, x_t^i) = \frac{p(x_{t-\Delta t}^i)}{p(x_t^i)}\boldsymbol{Q}_t^{\text{tok}}(x_t^i, x_{t-\Delta t}^i)$ and $x_{t-\Delta t}^i \neq x_t^i$, or $\bar{\boldsymbol{Q}}_t(x_{t-\Delta t}^i, x_t^i) = -\sum_{z \neq x_t} \bar{\boldsymbol{Q}}_t(z, x_t^i)$. The reverse equation is formulated as follows:

$$p(x_{t-\Delta t}^i | x_t^i) = \delta_{x_{t-\Delta t}^i x_t^i} + \bar{\boldsymbol{Q}}_t(x_{t-\Delta t}^i, x_t^i)\Delta t + o(\Delta t), \quad (2)$$

where we can estimate the ratio $\frac{p(x_{t-\Delta t}^i)}{p(x_t^i)}$ (which is known as the *concrete score* (Lou et al., 2024; Meng et al., 2022) to measure the *transition probability or closeness* from a state $x^i$ at time $t$ to a state $\hat{x}^i$ at time $t - \Delta t$) of $\bar{\boldsymbol{Q}}_t$ by a score network $s_\theta(x_t^i, t)_{x_{t-\Delta t}^i} \approx [\frac{p(x_{t-\Delta t}^i)}{p(x_t^i)}]_{x_t^i \neq x_{t-\Delta t}^i}$. So that the reverse matrix is parameterized to model the reverse process $q_\theta(x_{t-\Delta t}^i | x_t^i)$ (*i.e.* parameterize the concrete score).

**Training objective.** Denoising score entropy (DSE) (Lou et al., 2024) is introduced to train the score network $s_\theta$:

$$\int_0^T \mathbb{E}_{\boldsymbol{x}_t \sim p(\boldsymbol{x}_t | \boldsymbol{x}_0)} \sum_{\hat{\boldsymbol{x}}_t \neq \boldsymbol{x}_t} \boldsymbol{Q}_t\left(\hat{x}_t^i, x_t^i\right) \left[s_\theta\left(x_t^i, t\right)_{\hat{x}_t^i}\right.$$
$$\left. -c_{\hat{x}_t^i x_t^i} \log s_\theta\left(x_t^i, t\right)_{\hat{x}_t^i} + \text{N}(c_{\hat{x}_t^i x_t^i})\right] dt, \quad (3)$$

where the concrete score $c_{\hat{x}_t^i x_t^i} = \frac{p(\hat{x}_t^i | x_0^i)}{p(x_t^i | x_0^i)}$ and a normalizing constant function $\text{N}(c) := c \log c - c$ that ensures loss non-negative. During sampling, we can replace the concrete score with the trained score network on Equation (2).

# 4. Methodology

## 4.1. Task Formulation for eF2S

Given a triplet of multimodal-driven conditions $\boldsymbol{c} = \{\boldsymbol{c}_{\text{id}}, \boldsymbol{c}_{\text{emo}}, \boldsymbol{c}_{\text{text}}\}$, which correspond to reference identity, emotion, and text, respectively, the eF2S task aims to synthesize speech based on the $\boldsymbol{c}$. More precisely, the synthesized speech content aligns with the text condition $\boldsymbol{c}_{\text{text}}$, while its voice identity and emotional attributes correspond to the identity condition $\boldsymbol{c}_{\text{id}}$ and emotion condition $\boldsymbol{c}_{\text{emo}}$, respectively—both extracted from the input face.

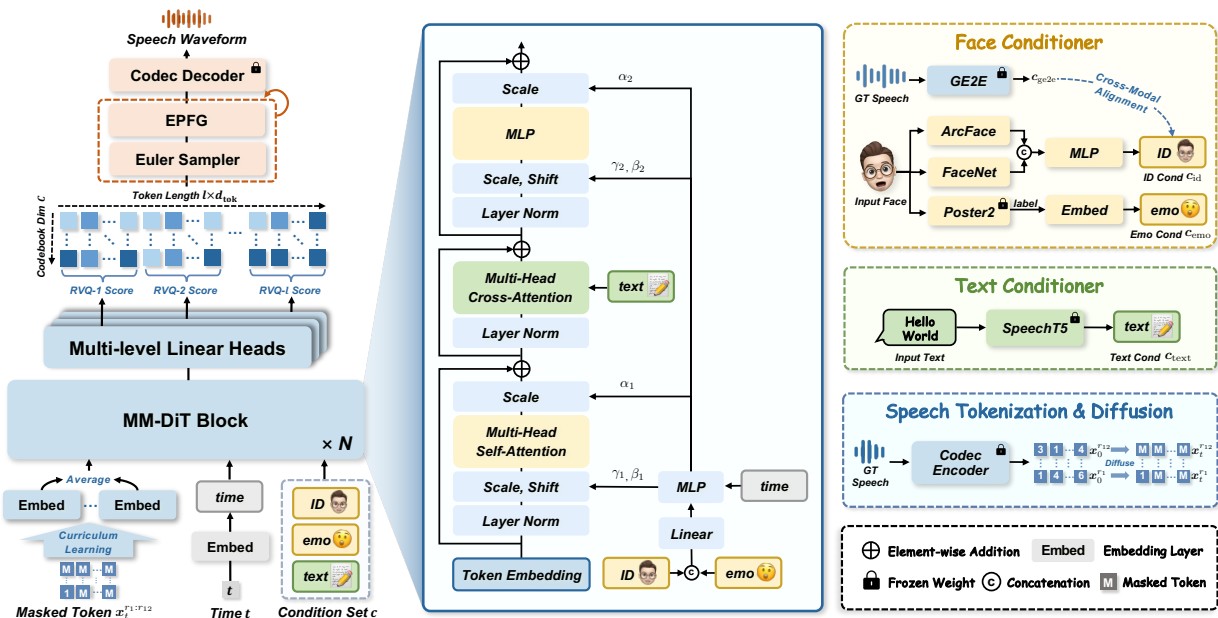

Figure 2: **Overall framework of DEmoFace**. The MM-DiT inputs masked token $x_t^{r_1:r_{12}}$, time $t$, and condition set $c$ to synthesize speech, consisting of $N$ blocks for conditioning and 12 linear heads to predict concrete scores. During training, we propose a curriculum learning that first inputs low-level tokens and refines them by adding high-level tokens progressively. During sampling, an Euler sampler with our EPFG refines the tokens, while a codec decoder reconstructs the waveform.

## 4.2. Overview of DEmoFace

Fig. 2 illustrates the overview of DEmoFace. The MM-DiT comprises $N$ blocks for conditional information injection and 12 linear heads for concrete score prediction. The masked tokens $x_t^{r_1:r_{12}}$ are obtained via speech tokenization and forward diffusion, with face and text conditioners forming the condition set $c$. Meanwhile, identity and emotion conditions with time are injected through adaptive layer normalization (AdaLN) (Peebles & Xie, 2023), and the text condition is injected with cross-attention. During training, we propose a curriculum learning algorithm, which first inputs low-level tokens $x_t^{r_1:r_{l-1}}$ and refines them by adding high-level token $x_t^{r_l}$ progressively. During sampling, we utilize an Euler sampler with our EPFG to iteratively refine the generated tokens, while a codec decoder reconstructs the waveform. Notably, when ground truth speech is provided during training, the reference features $c_{\text{ge2e}}$ are extracted from the GE2E (Wan et al., 2018) to guide identity customization. During inference, we use the cross-modal aligned face encoder to extract the $c_{\text{id}}$ instead of $c_{\text{ge2e}}$.

Next, we detail the key components in DEmoFace.

## 4.3. Conditional Concrete Score Modeling

For the concrete score $s_\theta$ modeling, we first define the tokenization and forward diffusion processes, followed by a description of the conditioners and architecture, and conclude with the modulation of the concrete score using EPFG.

**RVQ speech tokenization.** We utilize the recent RVQ-based codec (Wang et al., 2024) as the tokenizer, which achieves hierarchical modeling of diverse information across different RVQ layers. Given a single-channel speech signal, the tokenizer compresses it to the output tokens $x^{r_1:r_{12}} = \{1, \ldots, C_{\text{code}}\}^{12 \times d_{\text{tok}}}$, where $r_i$ is the $i$-th RVQ level of token, $d_{\text{tok}}$ is the length of the token sequence, respectively. The number of RVQ layers is 12 with a codebook size $C_{\text{code}} = 1,024$ in each layer.

**RVQ token diffusion process.** Given the hierarchical structure of RVQ tokens, following the previous diffusion process (Lou et al., 2024), we randomly corrupt each level token $x_t^{r_i}$ at timestep $t$. Specifically, we first extract input tokens $x_0^{r_1:r_l}$ from the codec encoder according to the curriculum training stage, where $r_l$ denotes the max level for the current input. We then conduct the diffusion process as defined in Equation (1) for $x_t^{r_i}$, where $1 \le i \le l$.

**Conditioners.** For face conditioner, as presented in Fig. 2, we build identity encoder and emotion encoder to learn identity embedding $c_{\text{id}}$ and emotion embedding $c_{\text{emo}}$, respectively. Specifically, we first employ a composite identity embedding by introducing two face recognition models ArcFace (Deng et al., 2022) and FaceNet (Schroff et al., 2015), then utilize a multilayer perceptron (MLP) for transformation and shape alignment. To precisely model the high-fidelity vocal style associated with the face, we extract the speech speaker embedding $c_{\text{ge2e}}$ from the speaker

recognition model GE2E (Wan et al., 2018), and make $c_{id}$ aligned with the $c_{ge2e}$ across modalities using cosine similarity, L1, and L2 losses, as detailed in Sec. 4.4. For the emotion embedding, we employ a strong facial expression recognition model Poster2 (Mao et al., 2023). Since the continuous emotion embedding of backbone is insufficient for decoupling identity information (Liu et al., 2024), we leverage the predicted label and a learnable embedding layer to learn identity-agnostic embedding $c_{emo}$.

For text conditioner, we introduce a text encoder to learn text embedding $c_{text}$. Specifically, raw text is preprocessed into an International Phonetic Alphabet (IPA) phone sequence using a standard IPA phonemizer. Next, embedding is extracted from a pre-trained text-speech encoder SpeechT5 (Ao et al., 2022), and is then subsequently projected into the hidden state via an MLP.

**Multimodal DiT.** We propose the Multimodal DiT (MM-DiT), which differs from DiT (Peebles & Xie, 2023) in three aspects. (1) *Input*, the masked speech tokens $x_t^{r_1:r_{12}}$ at timestep $t$ are fed to embedding layers and subsequently averaged to serve as the input. (2) *Conditioning*, to customize face-style speech generation, we concatenate $c_{id}$ and $c_{emo}$ along with the timestep embedding, is passed through an MLP to inject the global face-style condition. The MLP aims to regress the scale and shift parameters $\alpha_1, \gamma_1, \beta_1, \alpha_2, \gamma_2, \beta_2$ for the AdaLN. Additionally, to learn face-style linguistic expressions, we apply cross-attention with rotary position embeddings (Su et al., 2024) enabling dynamic alignment with text $c_{text}$. (3) *Output*, we incorporate 12 linear heads including a combination of AdaLN and linear layer to predict concrete scores for each RVQ level.

**Enhanced predictor-free guidance.** Several guidance tricks can boost sampling quality for the conditional generation, such as predictor-free guidance (PFG) (Nisonoff et al., 2024; Ho & Salimans, 2021). However, given $K$ conditions $c = \{c_1, \ldots, c_K\}$, these guidance methods are not readily amenable to multi-conditional scenarios (Liu et al., 2022). From the perspective of Energy-Based Models (EBMs), we propose an Enhanced PFG (EPFG) enhancing the efficient response to global condition while facilitating the decoupling of local conditions.

Specifically, to simplify the notation, we define $x_t^i, x_{t-\Delta t}^i$ as $x, \hat{x}$. The key of conditional sampling process is to estimate the concrete score $\hat{s}_\theta(x, t, c)_{\hat{x}} \approx \frac{p(\hat{x})}{p(x)}$ linked to the transition probability. Using Bayes rule, we can obtain a compositional concrete score $\hat{s}_\theta(x, t, c)_{\hat{x}}$ from $p(x_{t-\Delta t} = \hat{x}|x_t = x, c)$ based on Equation (2):

$$\hat{s}_\theta(x, t, c)_{\hat{x}} = s_\theta(x, t)_{\hat{x}} \prod_{k=1}^{K} \frac{s_\theta(x, t, c_k)_{\hat{x}}}{s_\theta(x, t)_{\hat{x}}}. \quad (4)$$

Instead of sampling from $\hat{s}_\theta(x, t, c)_{\hat{x}}$, we can utilize temperature sampling (Kingma & Dhariwal, 2018; Mehta et al., 2023) for more controllable generated outputs by introducing $\hat{s}_\theta^{(w)}(x, t, c)_{\hat{x}} = s_\theta(x, t)_{\hat{x}} \prod_{k=1}^{K} \frac{s_\theta^{w_k}(x,t,c_k)_{\hat{x}}}{s_\theta^{w_k}(x,t)_{\hat{x}}}$, where $w_k$ denotes the guidance scale. However, this compositional guidance lacks interactions among local conditions and struggles to guide sampling with a global consistent direction. Inspired by the formulation of the EBM, the score can also be formulated as $\hat{s}_\theta(x)_{\hat{x}} \approx \frac{p_\theta(\hat{x})}{p_\theta(x)} = \frac{e^{f_\theta(\hat{x})}/Z}{e^{f_\theta(x)}/Z} = \frac{e^{f_\theta(\hat{x})}}{e^{f_\theta(x)}}$, where $Z$ is the normalizing constant, and $f_\theta$ is the energy function. Hence, we can associate compositional and joint conditions by summing up the energy functions, and finally obtain the modulated score by multiplying both scores:

$$\hat{s}_\theta^{(w)}(x, t)_{\hat{x}} = s_\theta(x, t)_{\hat{x}} \underbrace{\prod_{k=1}^{K} \frac{s_\theta^{w_i}(x,t,c_k)_{\hat{x}}}{s_\theta^{w_i}(x,t)_{\hat{x}}}}_{\text{Compositional}} \cdot \underbrace{\frac{s_\theta^{w_0}(x,t,c)_{\hat{x}}}{s_\theta^{w_0-1}(x,t)_{\hat{x}}}}_{\text{Joint}}, \quad (5)$$

where $c = \{c_{id}, c_{emo}, c_{text}\}$, $w_0$ controls the scale of guidance strength for the joint injection of all conditions, while $w_i$ for $1 \le k \le K$ is assigned to each independent attribute. Please refer to Appendix C for detailed derivation.

### 4.4. Curriculum-based Training and Inference

**Training.** Curriculum learning aims to progressively train the model from simple to hard tasks, with the key challenge of identifying samples varying in difficulty. Previous studies show that neural networks prioritize low-frequency information first (Rahaman et al., 2019). Fig. 5(a) shows different frequency distributions across RVQ levels, with low-level features exhibiting low-frequency patterns. Therefore, we reveal curriculum learning for RVQ-based tokens from the frequency perspective. As shown in Fig. 2, we gradually introduce higher-level tokens $x_t^{r_l}$ every 3 epochs, starting from previous low-level (*i.e.* low-frequency) tokens $x_t^{r_1:r_{l-1}}$ to high-level tokens $x_t^{r_1:r_l}$, facilitating effective training.

Furthermore, the training procedure contains two stages: concrete score prediction and identity feature alignment. In the concrete score prediction, the training objective is the multi-level DSE loss based on Equation (3) with the sum across current $r_l$ RVQ levels as $\mathcal{L}_{score} = \sum_{i=1}^{l} \mathcal{L}_{DSE}(x^{r_i}, t, c)$. For conducting EPFG in Equation (5), we randomly set $\varnothing$ with 10% probability for each condition, and enforce all conditions set to $\varnothing$ for 10% samples. For the cross-modal alignment, we use cosine similarity, L1, and L2 losses to align the visual identity vectors with $c_{ge2e}$. With these compositional losses, the training objective for the face encoder is as $\mathcal{L}_{align} = 1 - \cos(c_{id}, c_{ge2e}) + L1(c_{id}, c_{ge2e}) + L2(c_{id}, c_{ge2e})$. Notably, to avoid information degradation with teacher-student distillation, we directly train MM-DiT with ground truth $c_{ge2e}$ and use the aligned identity embedding $c_{id}$ during inference.

| Methods | Audio | Visual | EmoSim↑ | SpkSim↑ | RMSE↓ | MCD↓ | WER(%)↓ |
|---|---|---|---|---|---|---|---|
| Ground Truth | - | - | 1.0000 | 1.0000 | 0.00 | 0.0000 | 10.82 |
| *Acoustic-guided Speech Generation* | | | | | | | |
| EmoSpeech (Diatlova & Shutov, 2023) | ✓ | ✗ | 0.7667 | 0.5677 | 114.70 | 7.1328 | 29.59 |
| FastSpeech2 (Ren et al., 2021) | ✓ | ✓ | 0.7010 | 0.5217 | 115.97 | 7.3461 | 29.49 |
| V2C-Net (Cong et al., 2023) | ✓ | ✓ | 0.6788 | 0.5773 | 115.55 | 6.8901 | 29.54 |
| HPM (Chen et al., 2022) | ✓ | ✓ | 0.6817 | 0.4404 | 97.19 | 7.7614 | 77.31 |
| StyleDubber (Cong et al., 2024b) | ✓ | ✓ | 0.6742 | 0.4753 | 103.59 | 7.4497 | 43.14 |
| DEMOFACE* (Ours) | ✓ | ✓ | **0.7921** | **0.7990** | **94.68** | **6.5505** | **19.72** |
| *Visual-guided Speech Generation* | | | | | | | |
| Face-TTS (Lee et al., 2023) | ✗ | ✓ | 0.5230 | 0.1968 | 118.96 | 8.4649 | **17.47** |
| DEmoFace (Ours) | ✗ | ✓ | **0.6965** | **0.6679** | **101.18** | **6.8601** | 20.78 |

Table 1: **Speech quantitative results.** The *Audio* and *Visual* indicate whether specific modality conditions are used for speech generation guidance. ↑ (↓) means the higher (lower) value is better. We **bold** the best-performing method. Notably, the * denotes that we use the speech condition $c_{\text{ge2e}}$, rather than the face condition $c_{\text{id}}$ to guide identity conditioning.

**Inference.** During inference, we introduce a frame-level duration predictor to estimate speech durations, initializing the input length $d_{\text{tok}}$. Then the reverse process is executed with Euler sampling (Lou et al., 2024) and EPFG with 96 steps. The details of the duration predictor refer to Appendix E.1.

## 5. Experimental Results

### 5.1. Experimental Setups

**Datasets.** All our models are pre-trained on three datasets with pairs of face video and speech: RAVDESS (Livingstone & Russo, 2018), MEAD (Wang et al., 2020; Gan et al., 2023), and MELD-FAIR (Carneiro et al., 2023). For data pre-processing, we first resample the audio to 16 kHz, and apply a speech separation model SepFormer (Subakan et al., 2021) to enhance voice. Then, we introduce Whisper (Radford et al., 2023) to filter non-aligned text-speech pairs. Then, all models are trained on a combination of all three datasets. The RAVDESS and MEAD of the combined one are randomly segmented into training, validation, and test sets without any speaker overlap. For the MELD-FAIR, we follow the original splits. Additionally, these datasets lack sufficient semantic units in real-world environments, making it challenging to train a TTS model. We incorporate a 10-hour subset from LRS3 (Afouras et al., 2018) for pre-training, allowing the model to be comparable to Face-TTS trained on 400 hours of LRS3. Finally, the combined dataset comprises 31.33 hours of audio recordings and 26,767 utterances across 7 basic emotions (*i.e.* angry, disgust, fear, happy, neutral, sad, and surprised) and 953 speakers.

**Evaluation metrics.** For eF2S, we evaluate the generation performance based on naturalness (*i.e.* speech quality) and expressiveness. For the naturalness, we employ Mel

Cepstral Distortion (MCD) (Chen et al., 2022) to assess discrepancies between generated and target speech. Additionally, the Word Error Rate (WER) (Wang et al., 2018; Radford et al., 2023) is used to gauge intelligibility. For the expressiveness, we calculate cosine similarity metrics based on emotion embeddings (Ma et al., 2024) and d-vectors to assess emotion similarity (EmoSim) and speaker identity similarity (SpkSim), as well as the Root Mean Square Error (RMSE) for F0 (Hayashi et al., 2017).

**Implementation details.** We implement DEmoFace based on DiT architecture (Peebles & Xie, 2023). We use a log-linear noise schedule $\sigma(t)$ (Lou et al., 2024) where the expectation of the number of masked tokens is linear with $t$. During training, we use the AdamW optimizer (Loshchilov & Hutter, 2019) with a learning rate of 1e-4, batch size 32. The total number of iterations is 300k. During inference, we use the Euler sampler to conduct the reverse process with 96 steps. We set the joint guidance scale $w_0 = 1.9$, and compositional scales $w_1 = w_2 = 1.0$, $w_3 = 1.6$. Please refer to Appendix E for more implementation details.

### 5.2. Quantitative Evaluation

For quantitative evaluation, we compare DEmoFace with previous state-of-the-art (SOTA) methods, categorized into two paradigms based on the type of guidance. The acoustic-guided methods customize identity using acoustic prompts, and drive emotion generation from either visual or acoustic cues, while the visual-guided methods aim to customize both identity and emotion only from visual conditions.

**Objective evaluation.** As shown in Tab. 1, compared with Face-TTS, we achieve 17.35% and 47.11% improvements in terms of EmoSim and SpkSim, reflecting the great abil-

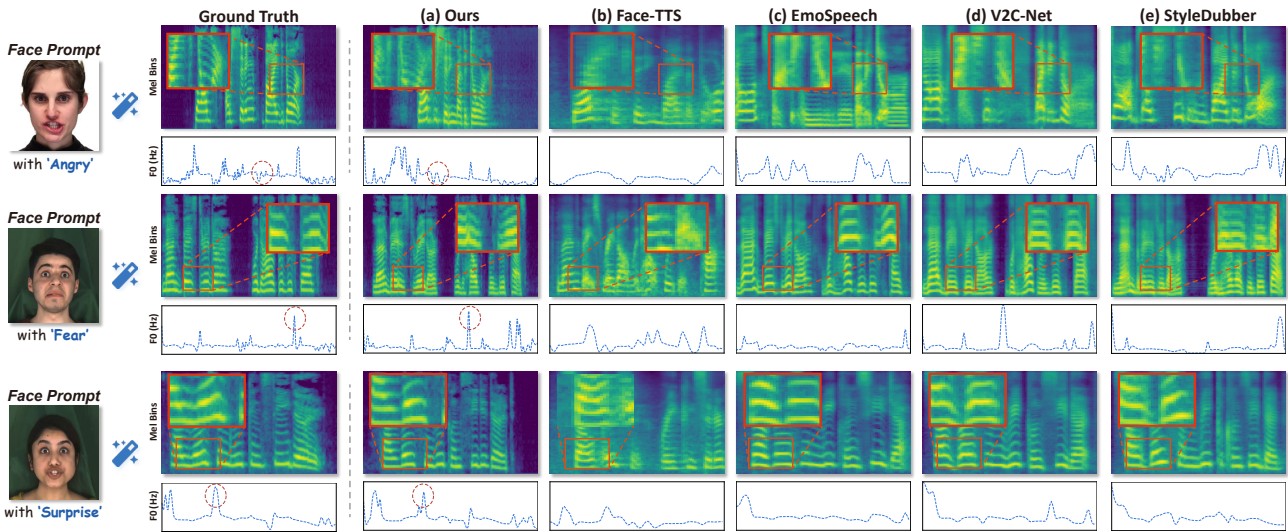

Figure 3: **Speech qualitative results.** The red rectangles highlight key regions with acoustic differences or over-smoothing issues, and the red dotted circle shows similar F0 contours with ground truth. Zoom in for more details.

ity to maintain voice-identity while enhancing consistency. For prosody modeling, we can estimate a more precise F0 contour with relative 14.95% gains, exhibiting more natural speech expressiveness. The MCD improved by a relative 18.96%, indicating minimal acoustic difference with the target speech. While Face-TTS achieves a better WER by utilizing over 10 times the data, DEmoFace significantly improves naturalness and consistency with fewer data.

Notably, we observe that the visual-guided DEmoFace even outperforms the acoustic-guided methods, which are the most efficient for speech generation by leveraging isomorphic features. It demonstrates that DEmoFace bridges the cross-modal gap using only heterogeneous face features. Furthermore, we introduce the acoustic-guided DEMO-FACE* replacing face condition $c_{id}$ with speech condition $c_{ge2e}$, which gains greater improvements than other acoustic-guided methods in all metrics by a large margin.

**Subjective evaluation.** We further conduct the subjective evaluation with 50 participants, to compare our DEmoFace with SOTA methods (Diatlova & Shutov, 2023; Lee et al., 2023). Specifically, we introduce three metrics based on five mean opinion scores (MOS), each rated on a scale from 1 to 5 in 0.5 increments. Among them, $MOS_{nat}$ evaluates speech naturalness and generation quality, while $MOS_{id}$ and $MOS_{emo}$ assess the perceived similarity of timbre and emotional prosody, respectively. We randomly generate 15 utterances from the test set for subjective evaluation.

The scoring results of the user study, summarized with 95% confidence intervals in Tab. 2, demonstrate that DEmoFace has a clear advantage over SOTA methods in all metrics.

| Methods | $MOS_{nat}$ ↑ | $MOS_{id}$ ↑ | $MOS_{emo}$ ↑ |
|---|---|---|---|
| EmoSpeech | 2.30±0.19 | 2.93±0.09 | 2.78±0.13 |
| Face-TTS | 2.28±0.09 | 2.67±0.12 | 2.75±0.09 |
| DEmoFace (Ours) | **3.17±0.18** | **3.20±0.11** | **3.26±0.12** |

Table 2: **Subjective evaluation** on speech naturalness, and similarity of identity (*i.e.* timbre) and emotion, compared with acoustic- and visual-guided methods.

Notably, it achieves a 38% relative improvement in $MOS_{nat}$, confirming its superiority in generating high-quality speech. Furthermore, compared to the acoustic-guided EmoSpeech, DEmoFace yields higher $MOS_{emo}$ and $MOS_{id}$ scores, reflecting its ability to generate speech with greater emotional and identity consistency.

### 5.3. Qualitative Results

**Qualitative comparisons.** As shown in Fig. 3, from mel-spectrogram in the first row, we observe severe temporal inconsistencies and over-smoothing issues in Face-TTS, EmoSpeech, and V2C-Net, which result in misaligned speech durations and a significant degradation in generation quality. Furthermore, the F0 curves in the third row indicate that these baselines produce pitch contours that deviate markedly from the ground truth (GT), reflecting mismatches in prosodic features such as pitch height, emotion, and intonational patterns. In contrast, our DEmoFace produces speeches that are more closely aligned with the GT in both spectrogram structure and F0 dynamics, highlighting the benefits of our enhanced multi-conditional generation and dynamic synchronization capabilities.

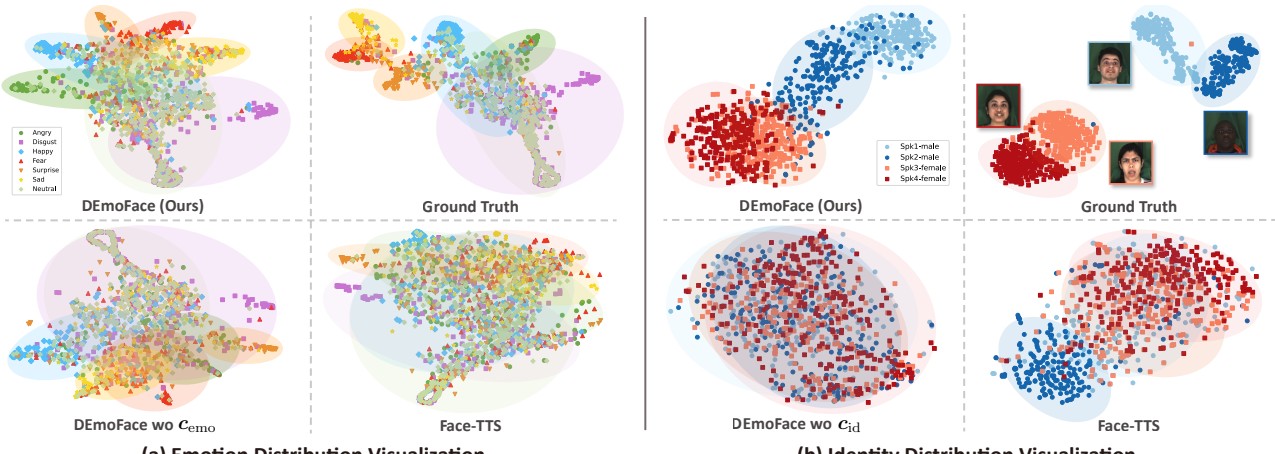

**(a) Emotion Distribution Visualization**   **(b) Identity Distribution Visualization**

Figure 4: **t-SNE visualization** based on emotion2vec (*i.e.* speech emotion features) and d-vector (*i.e.* speech identity features) representations. Each point represents a speech sample, and colors indicate different emotions or identities. Points in close distance indicate higher similarity, while overlapping points suggest potential confusion. We compare our DEmoFace with real speech, an ablation without emotional or identity conditioning, and Face-TTS baseline. Results show that we achieve better clustering than other methods with great ability in emotional and identity generation.

**Visualization of speaker embeddings.** To explore the diversity of emotional expressions and speaker identities in the generated speech, we utilize the t-SNE technique (Van der Maaten & Hinton, 2008) to visualize the distribution of emotion2vec features and d-vector features extracted from the synthesized speech. As shown in Fig. 4, ablating the emotion condition $c_{emo}$ or speaker condition $c_{id}$ leads to wrong emotion or speaker clustering. The feature overlap implies that the model may struggle to generate speech with clearly distinguishable identity and affective characteristics, validating the effectiveness of our proposed emotional and identity conditioning. Furthermore, Face-TTS exhibits noticeable wrong mixing among different emotions and speakers, suggesting that it fails to perceive speaker timbre and emotional prosody effectively. In contrast, our DEmoFace effectively clusters the speech embeddings from the same speaker and emotion, while maintaining stronger speaker- and emotion-discriminative properties, indicating the ability to synthesize diverse vocal styles and faithful speeches from visual cues.

### 5.4. Ablation Studies

**Ablation on multimodal conditioning.** As the variant (a) in Tab. 3, the results demonstrate that incorporating $c_{id}$ significantly enhances speaker similarity with 26% SpkSim improvement and speech naturalness with 17% MCD relative improvement, while maintaining stable EmoSim and WER scores. Similarly, the inclusion of $c_{emo}$ leads to improved emotional prosody in EmoSim and RMSE without compromising intelligibility. These results indicate that DEmoFace can successfully decouple different conditioning signals from others through our EPFG algorithm.

| Vars | EmoSim↑ | SpkSim↑ | RMSE↓ | MCD↓ | WER↓ |
|------|---------|---------|-------|------|------|
| (a)  | 0.69    | 0.41    | 109.62 | 8.25 | 22.35 |
| (b)  | 0.64    | 0.65    | 104.92 | 7.29 | 21.35 |
| (c)  | 0.67    | 0.66    | 104.41 | 7.24 | 27.52 |
| (d)  | 0.69    | 0.58    | 115.30 | 8.33 | 39.06 |
| (e)  | 0.67    | 0.63    | 106.67 | 7.31 | 40.13 |
| Ours | **0.70** | **0.67** | **101.18** | **6.86** | **20.78** |

Table 3: **Ablation studies.** "Vars" refers to the ablation variants, with (a) to (e) indicating the removal of identity condition $c_{id}$, emotion condition $c_{emo}$, curriculum learning, identity feature alignment, and EPFG, respectively.

**Ablation on curriculum learning.** To demonstrate the effectiveness of curriculum learning, we input all RVQ-level tokens during the whole training process as the variant (c) in Tab. 3. Based on the fact that RVQ Codec preserves semantic information in low-level tokens while retaining acoustic details in high-level tokens (Nishimura et al., 2024), Fig. 3 shows: (1) we achieve better WER and EmoSim than baseline during early training, as prioritized low-level learning effectively captures low-level information; (2) SpkSim initially lags due to unseen high-level tokens but improves as they are progressively introduced. The results highlight the effectiveness of curriculum learning.

**Ablation on identity alignment.** Due to the heterogeneous differences between speaker features extracted from vision and speech, achieving accurate cross-modal identity customization remains challenging. As shown in Tab. 3 (d), removing identity alignment leads to a significant 9% drop in SpkSim, indicating a clear misalignment between real

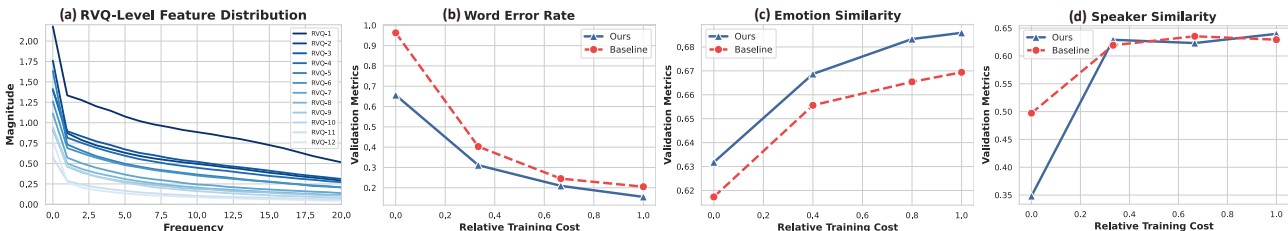

Figure 5: **Ablation study on curriculum learning**. (a) Feature distribution across RVQ levels, with low-level features showing low-frequency patterns. (b)-(d) For the baseline without curriculum learning, we vary the number of training epochs compared with three metrics on the validation set. The effect is evident for WER and EmoSim while slight on SpkSim.

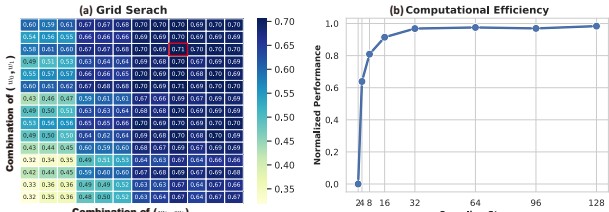

Figure 6: (a) Parameters grid search for the EPFG, with axes as two-parameter combinations, colors as normalized performance. (b) Effect on sampling steps.

timbre and visual identity embedding. Moreover, the presence of noise in visual features further disrupts the generation process, resulting in degraded performance across all evaluation metrics.

**Ablation on EPFG.** The EPFG enables an effective generalization across combinations of multiple conditions, even those unseen during pre-training. From Tab. 3 (e), we observe that EmoSim, SpkSim, and WER metrics have degraded when all conditions are treated as a unified one, showing the incorporation of EPFG can significantly enhance multi-conditional generation quality. Furthermore, we conduct a grid search for all parameters of the EPFG on the validation set. As shown in Fig. 6(a), axes denote two parameter combinations (*i.e.* $(w_0 \in [1.0, 2.0], w_1 \in [1.0, 1.4])$, $(w_1 \in [1.0, 1.4], w_2 \in [1.0, 2.0])$), the color of the grid indicates normalized performance score, and the red rectangle marking the final parameter combination we select. We observe performance degradation (*i.e.* the light area) with complex entanglement, as the unconditional score dominates with low guidance scales across conditions, further demonstrating the effectiveness of the EPFG.

**Ablation on inference efficiency.** To explore the effectiveness of sampling steps, we first normalize each metric to $[0, 1]$ and obtain the average performance. As shown in Fig. 6(b), the performance improves with more steps, saturating at 32 steps. It demonstrates that DEmoFace achieves acceptable generation quality with just 32 steps. To balance

performance with efficiency, we utilize 96 steps in this paper. Furthermore, we evaluate inference Real-Time Factor (RTF) to measure the time (in seconds) required to synthesize one second of speech. Specifically, we utilize a mini-batch size of 1 and 32 with the Number of Function Evaluations (NFE) fixed at 32. When using a batch size of 1, our method achieves an RTF of 0.49, demonstrating the potential for real-time speech synthesis—approximately twice as fast as real-time. With a batch size of 32, the RTF further improves to 0.13, highlighting the efficiency of DEmoFace.

## 6. Conclusion

We propose DEmoFace, the first RVQ-based discrete diffusion framework for eF2S with high diversity and quality, serving as a foundation for future research in multimodal personalized TTS systems. Specifically, the MM-DiT dynamically aligns speech and text prompts, and customizes face-style linguistic expressions. Meanwhile, we introduce a coarse-to-fine curriculum learning strategy to enhance training efficiency. Moreover, to boost efficient response to the global condition while facilitating the decoupling of multiple local conditions, we develop an enhanced predictor-free guidance to achieve multi-conditional generation of eF2S. Extensive experimental results show that DEmoFace outperforms existing methods and effectively synthesizes the speech solely from the visual cues.

**Limitation.** Despite achieving advanced performance, we struggle to precisely reconstruct a true voice due to visual-voice biases within the dataset, tending to produce average-sounding speech. Furthermore, the limited size of the dataset poses a significant challenge to achieving accurate pronunciation. We will incorporate larger datasets to improve the diversity and intelligibility of the model.

## Acknowledgment

This work was supported in part by STI2030-Major Projects (No. 2021ZD0200204), National Natural Science Foundation of China (No. 62471148), and Shanghai Center for Brain Science and Brain-inspired Technology.

## Impact Statement

Given the privacy of identity information in face and speech, ensuring compliance with ethical standards and privacy rules is essential to safeguard individual rights and prevent potential misuse. We stress the necessity of consent agreements for using published models, to ensure responsible application while respecting individual privacy and rights.

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

# Appendix

This appendix provides the following extra contents:

- Appendix A shows detailed notations and definitions;

- Appendix B provides a preliminary of the discrete diffusion model;

- Appendix C presents a detailed derivation of enhanced predictor-free guidance;

- Appendix D includes the statics of datasets used in this paper;

- Appendix E supplements the experimental details of our DEmoFace and each baseline method;

- Appendix F contains extra experimental results; and

- Appendix G incorporates the details of subjective evaluation.

## A. Detailed Notations and Definitions

Tab. A-1 provides the notations and definitions of variables used in the paper.

| Notation | Definition |
|---:|:---|
| $\boldsymbol{x}$ | Vector variables representing a sequence of tokens. |
| $x^i, \hat{x}, x_t^i, \hat{x}_t^i$ | Scalar variables representing token or state in discrete diffusion process. |
| $\mathcal{X}^d$ | State space $\{1, \ldots, n\}^d$ of token sequence length $d$ and token dimension $n$. |
| $\boldsymbol{Q}_t$ | Diffusion forward matrix (*i.e.* transition rate matrix) at $t$ time. |
| $\bar{\boldsymbol{Q}}_t$ | Diffusion reverse matrix at $t$ time. |
| $\boldsymbol{Q}^{\text{tok}}$ | Token-level transition rate matrix filled with absorbing state [MASK]. |
| $\delta_{x\hat{x}}$ | Kronecker delta function of two variables $x, \hat{x}$, which is 1 if the variables are equal, and 0 otherwise. |
| $p$ | Probability distribution of the forward diffusion process characterized by $\boldsymbol{Q}_t$. |
| $q_\theta$ | Probability distribution of the reverse diffusion process characterized by score model $s_\theta$. |
| $\boldsymbol{P}_{t\|0}$ | Transition probability matrix from time 0 to time $t$. |
| $s_\theta$ | Score network to estimate the ratio (*i.e.* concrete score) $\frac{p(x_{t-\Delta t}^i)}{p(x_t^i)}$. |
| $c_{\hat{x}_t x_t}$ | Concrete score $\frac{p(\hat{x}_t\|x_0)}{p(x_t\|x_0)}$. |
| $\mathrm{N}(c)$ | Normalizing constant function of denoising score entropy loss. |
| $r_i$ | $i$-th level of RVQ tokens, where $1 \leq i \leq 12$. $r_l$ denotes the current max level during our curriculum training. |
| $C_{\text{code}}$ | Codebook size. |
| $d_{\text{tok}}$ | Length of the token sequence. |
| $\boldsymbol{c}$ | Condition set including $\boldsymbol{c}_{\text{id}}, \boldsymbol{c}_{\text{emo}}, \boldsymbol{c}_{\text{text}}$. |
| $\alpha_1, \gamma_1, \beta_1$ | Scale and shift parameters for the adaptive layer normalization. |

Table A-1: Detailed Notations and Definitions

## B. Preliminary: Discrete Diffusion Models

Continuous Diffusion Models (CDM) have been one of the most prominent and active areas in generative modeling (Blattmann et al., 2023; Li et al., 2023; Ruan et al., 2023) , which has shown state-of-the-art performance in various fields. However, for speech generation, the high-dimensional speech features and excessive diffusion steps lead to high resource usage and inefficient inference, frustrating practical application. The fundamental way lies in compressing the speech feature space, such as a discrete space.

In recent years, Discrete diffusion models (DDM) have shown promise in language modeling (Austin et al., 2021; Meng et al., 2022; Lou et al., 2024; Ou et al., 2024), which are characterized by a forward and reverse process like continuous diffusion models (Song et al., 2021; Ho et al., 2020). Nevertheless, DDM has yet to be explored in speech generation. In

this study, we introduce the DDM for speech token generation. Below, we outline the forward and reverse processes, along with the training objective.

**Forward diffusion process.** Given a sequence of tokens $\boldsymbol{x} = x^1 \ldots x^d$ from a state space of length $d$ like $\mathcal{X}^d = \{1, \ldots, n\}^d$. The continuous-time discrete Markov chain at time $t$ is characterized by the diffusion matrix $\boldsymbol{Q}_t \in \mathbb{R}^{n^d \times n^d}$ (*i.e.* transition rate matrix), as follows:

$$p(x^i_{t+\Delta t}|x^i_t) = \delta_{x^i_{t+\Delta t} x^i_t} + \boldsymbol{Q}_t(x^i_{t+\Delta t}, x^i_t)\Delta t + o(\Delta t), \tag{A-1}$$

where $x^i_t$ denotes $i$-th element of $\boldsymbol{x}_t$, $\boldsymbol{Q}_t(x^i_{t+\Delta t}, x^i_t)$ is the $(x^i_{t+\Delta t}, x^i_t)$ element of $\boldsymbol{Q}_t$, denoting the transition rate from state $x^i_t$ to state $x^i_{t+\Delta t}$ at time $t$, and $\delta$ is the Kronecker delta. Since the exponential size of $\boldsymbol{Q}_t$, existing works (Lou et al., 2024; Ou et al., 2024) propose to assume dimensional independence, conducting a one-dimensional diffusion process for each dimension with the same token-level diffusion matrix $\boldsymbol{Q}^{\text{tok}}_t = \sigma(t)\boldsymbol{Q}^{\text{tok}} \in \mathbb{R}^{n \times n}$, where $\sigma(t)$ is the noise schedule and $\boldsymbol{Q}^{\text{tok}}$ is designed to diffuse towards an absorbing state [MASK] in this study. Then the forward equation is formulated as follows:

$$\boldsymbol{P}(x^i_t, x^i_0) = \exp\left(\bar{\sigma}(t)\boldsymbol{Q}^{\text{tok}}(x^i_t, x^i_0)\right), \tag{A-2}$$

where transition probability matrix $\boldsymbol{P}(x^i_t, x^i_0) := p(x^i_t|x_0)$, and cumulative noise $\bar{\sigma}(t) = \int_0^t \sigma(s)ds$. There are two probabilities in the $\boldsymbol{P}_{t|0}$: $1 - e^{-\bar{\sigma}(t)}$ for replacing the current tokens with [MASK], $e^{-\bar{\sigma}(t)}$ for keeping it unchanged, where the diffusion transition rate matrix $\boldsymbol{Q}^{\text{tok}}$ is defined as:

$$\boldsymbol{Q}^{\text{tok}} = \begin{bmatrix} -1 & 0 & \cdots & 0 & 1 \\ 0 & -1 & \cdots & 0 & 1 \\ \vdots & \vdots & \ddots & \vdots & \vdots \\ 0 & 0 & \cdots & -1 & 1 \\ 0 & 0 & \cdots & 0 & 0 \end{bmatrix}. \tag{A-3}$$

Therefore, we can parallel sample the corrupted sequence $\boldsymbol{x}_t$ directly from $\boldsymbol{x}_0$ in one step. During the inference, we start from $\boldsymbol{x}_T$ filled with [MASK] tokens and iteratively sample new set of tokens $\boldsymbol{x}_{t-1}$ from $p_\theta(\boldsymbol{x}_{t-1}|\boldsymbol{x}_t)$.

**Reverse denoising process.** As the transition rate matrix $\boldsymbol{Q}^{\text{tok}}_t$ is known, the reverse process can be given by a reverse transition rate matrix $\bar{\boldsymbol{Q}}_t$ (Sun et al., 2023; Kelly, 2011), where $\bar{\boldsymbol{Q}}_t(x^i_{t-\Delta t}, x^i_t) = \frac{p(x^i_{t-\Delta t})}{p(x^i_t)}\boldsymbol{Q}^{\text{tok}}_t(x^i_t, x^i_{t-\Delta t})$ and $x^i_{t-\Delta t} \neq x^i_t$, or $\bar{\boldsymbol{Q}}_t(x^i_{t-\Delta t}, x^i_t) = -\sum_{z \neq x_t} \bar{\boldsymbol{Q}}_t(z, x^i_t)$. The reverse equation is formulated as follows:

$$p(x^i_{t-\Delta t}|x^i_t) = \delta_{x^i_{t-\Delta t} x^i_t} + \bar{\boldsymbol{Q}}_t(x^i_{t-\Delta t}, x^i_t)\Delta t + o(\Delta t), \tag{A-4}$$

where we can estimate the ratio $\frac{p(x^i_{t-\Delta t})}{p(x^i_t)}$ (which is known as the *concrete score* (Lou et al., 2024; Meng et al., 2022) to measure the *transition probability or closeness* from a state $x^i$ at time $t$ to a state $\hat{x}^i$ at time $t - \Delta t$) of $\bar{\boldsymbol{Q}}_t$ by a score network $s_\theta(x^i_t, t)_{x^i_{t-\Delta t}} \approx [\frac{p(x^i_{t-\Delta t})}{p(x^i_t)}]_{x^i_t \neq x^i_{t-\Delta t}}$. So that the reverse matrix is parameterized to model the reverse process $q_\theta(x^i_{t-\Delta t}|x^i_t)$ (*i.e.* parameterize the concrete score).

**Training objective.** Denoising score entropy (DSE) (Lou et al., 2024) is introduced to train the score network $s_\theta$:

$$\int_0^T \mathbb{E}_{\boldsymbol{x}_t \sim p(\boldsymbol{x}_t|\boldsymbol{x}_0)} \sum_{\hat{\boldsymbol{x}}_t \neq \boldsymbol{x}_t} \boldsymbol{Q}_t\left(\hat{x}^i_t, x^i_t\right) \left[s_\theta\left(x^i_t, t\right)_{\hat{x}^i_t} - c_{\hat{x}^i_t x^i_t} \log s_\theta\left(x^i_t, t\right)_{\hat{x}^i_t} + \text{N}(c_{\hat{x}^i_t x^i_t})\right] dt, \tag{A-5}$$

where the concrete score $c_{\hat{x}^i x^i} = \frac{p(\hat{x}^i_t|x^i_0)}{p(x^i_t|x^i_0)}$ and a normalizing constant function $\text{N}(c) := c \log c - c$ that ensures loss non-negative. After training, we can replace the concrete score with the trained score network on Equation (A-4), conducting the sampling process.

## C. Derivation of Enhanced Predictor-free Guidance

For the discrete diffusion model, given the random variable value $\boldsymbol{x}_t = x_t^1 \ldots x_t^d$ from a state space of length $d$ with the absorbing state $\texttt{[MASK]}$ at time $t$, the *unconditional* probability distribution $p(x_{t-\Delta t}^i|x_t^i) = \delta_{x_{t-\Delta t}^i x_t^i} + \bar{\boldsymbol{Q}}_t(x_{t-\Delta t}^i, x_t^i)\Delta t + o(\Delta t)$ during sampling as Equation (A-4) shown. For the *conditional* probability distribution $p(x_{t-\Delta t}^i|x_t^i, \boldsymbol{c})$, the key is to obtain the conditional transition rate matrix $\bar{\boldsymbol{Q}}_t(x_{t-\Delta t}^i, x_t^i|\boldsymbol{c})$.

Firstly, following (Nisonoff et al., 2024) to simplify the notation, we define $x_t^i$ as $x_t$ and utilize the properties of the Kronecker delta $\delta$ (*i.e.* the function is 1 if the variables are equal, and 0 otherwise) to derive another form of the unconditional probability distribution $p(x_{t-\Delta t} = \hat{x}|x_t = x)$:

$$
\begin{aligned}
p(x_{t-\Delta t} = \hat{x}|x_t = x) &= \delta_{x_{t-\Delta t}=\hat{x},x_t=x} + \bar{\boldsymbol{Q}}_t(x_{t-\Delta t}=\hat{x}, x_t=x)\Delta t + o(\Delta t) \\
&= \delta_{\hat{x},x} + \delta_{\hat{x},x}\bar{\boldsymbol{Q}}_t(\hat{x},x)\Delta t + (1-\delta_{\hat{x},x})\bar{\boldsymbol{Q}}_t(\hat{x},x)\Delta t + o(\Delta t) \\
&= \delta_{\hat{x},x}\left(1 + \bar{\boldsymbol{Q}}_t(x,x)\Delta t\right) + (1-\delta_{\hat{x},x})\bar{\boldsymbol{Q}}_t(\hat{x},x)\Delta t + o(\Delta t).
\end{aligned}
\tag{A-6}
$$

Then, we utilize the formulation in Equation (A-6) and the Bayes rule to build the conditional probability distribution $p(x_{t-\Delta t} = \hat{x}|x_t = x, \boldsymbol{c})$, combining predictive distribution $p(\boldsymbol{c}|x)$ and unconditional distribution $p(x_{t-\Delta t} = \hat{x}|x_t = x)$ as:

$$
\begin{aligned}
p(x_{t-\Delta t} = \hat{x}|x_t = x, \boldsymbol{c}) &= \frac{p(\boldsymbol{c}|x_{t-\Delta t} = \hat{x}, x_t = x)p(x_{t-\Delta t} = \hat{x}|x_t = x)}{p(\boldsymbol{c}|x_t = x)} \\
&= \frac{p(\boldsymbol{c}|x_{t-\Delta t} = \hat{x}, x_t = x)p(x_{t-\Delta t} = \hat{x}|x_t = x)}{\sum_{x'} p(\boldsymbol{c}|x_{t-\Delta t} = x', x_t = x)p(x_{t-\Delta t} = x'|x_t = x)} \\
&= \frac{\frac{p(\boldsymbol{c}|x_{t-\Delta t}=\hat{x},x_t=x)}{p(\boldsymbol{c}|x_{t-\Delta t}=x,x_t=x)}\left[\delta_{\hat{x},x}\left(1 + \bar{\boldsymbol{Q}}_t(x,x)\Delta t\right) + (1-\delta_{\hat{x},x})\bar{\boldsymbol{Q}}_t(\hat{x},x)\Delta t + o(\Delta t)\right]}{\sum_{x'}\frac{p(\boldsymbol{c}|x_{t-\Delta t}=x',x_t=x)}{p(\boldsymbol{c}|x_{t-\Delta t}=x,x_t=x)}\left[\delta_{\hat{x},x}\left(1 + \bar{\boldsymbol{Q}}_t(x,x)\Delta t\right) + (1-\delta_{x',x})\bar{\boldsymbol{Q}}_t(x',x)\Delta t + o(\Delta t)\right]},
\end{aligned}
\tag{A-7}
$$

where we use Equation (A-6) to replace $p(x_{t-\Delta t} = \hat{x}|x_t = x)$ in A-7 and define $p_c(\hat{x}, x) \equiv p(\boldsymbol{c}|x_{t-\Delta t} = \hat{x}, x_t = x)$. We further simplify the formulation:

$$
\begin{aligned}
p(x_{t-\Delta t} = \hat{x}|x_t = x, \boldsymbol{c}) &= \frac{\frac{p_c(\hat{x},x)}{p_c(x,x)}\left[\delta_{\hat{x},x}\left(1 + \bar{\boldsymbol{Q}}_t(x,x)\Delta t\right) + (1-\delta_{\hat{x},x})\bar{\boldsymbol{Q}}_t(\hat{x},x)\Delta t + o(\Delta t)\right]}{\sum_{x'}\frac{p_c(x',x)}{p_c(x,x)}\left[\delta_{\hat{x},x}\left(1 + \bar{\boldsymbol{Q}}_t(x,x)\Delta t\right) + (1-\delta_{x',x})\bar{\boldsymbol{Q}}_t(x',x)\Delta t + o(\Delta t)\right]} \\
&= \frac{\delta_{\hat{x},x}\left(1 + \bar{\boldsymbol{Q}}_t(x,x)\Delta t\right) + \frac{p_c(\hat{x},x)}{p_c(x,x)}(1-\delta_{\hat{x},x})\bar{\boldsymbol{Q}}_t(\hat{x},x)\Delta t + o(\Delta t)}{\left(1 + \bar{\boldsymbol{Q}}_t(x,x)\Delta t\right) + \sum_{x'\neq x}\frac{p_c(x',x)}{p_c(x,x)}\bar{\boldsymbol{Q}}_t(x',x)\Delta t + o(\Delta t)} \\
&= \frac{\delta_{\hat{x},x}\left(1 + \bar{\boldsymbol{Q}}_t(x,x)\Delta t\right) + \frac{p_c(\hat{x},x)}{p_c(x,x)}(1-\delta_{\hat{x},x})\bar{\boldsymbol{Q}}_t(\hat{x},x)\Delta t + o(\Delta t)}{1 + f(\Delta t, x, x')},
\end{aligned}
\tag{A-8}
$$

where $f$ is a function of $\Delta t$, and as $\Delta t \to 0$ we can use Taylor expansion of $\frac{1}{1+f(\Delta t,x,x')} \approx 1 - f(\Delta t, x, x') + o(\Delta t^2)$:

$$
\begin{aligned}
p(x_{t-\Delta t} = \hat{x}|x_t = x, \boldsymbol{c}) =& \left[\delta_{\hat{x},x}\left(1 + \bar{\boldsymbol{Q}}_t(x,x)\Delta t\right) + \frac{p_c(\hat{x},x)}{p_c(x,x)}(1-\delta_{\hat{x},x})\bar{\boldsymbol{Q}}_t(\hat{x},x)\Delta t + o(\Delta t)\right] \\
&\times \left[1 - \bar{\boldsymbol{Q}}_t(x,x)\Delta t - \sum_{x'\neq x}\frac{p_c(x',x)}{p_c(x,x)}\bar{\boldsymbol{Q}}_t(x',x)\Delta t + o(\Delta t)\right] \\
=& \delta_{\hat{x},x}\left(1 - \sum_{x'\neq x}\frac{p_c(x',x)}{p_c(x,x)}\bar{\boldsymbol{Q}}_t(x',x)\Delta t\right) + (1-\delta_{\hat{x},x})\frac{p_c(\hat{x},x)}{p_c(x,x)}\bar{\boldsymbol{Q}}_t(\hat{x},x)\Delta t + o(\Delta t).
\end{aligned}
\tag{A-9}
$$

From the expression in Equation (A-6) and property of the $\bar{\boldsymbol{Q}}$ (*i.e.* $\bar{\boldsymbol{Q}}_t(x,x) + \sum_{x'\neq x}\bar{\boldsymbol{Q}}_t(x',x) = 0$) (Lou et al., 2024), we can derive our conditional transition rate matrix:

$$
\bar{\boldsymbol{Q}}_t(\hat{x}, x|\boldsymbol{c}) = \frac{p_c(\hat{x},x)}{p_c(x,x)}\bar{\boldsymbol{Q}}_t(\hat{x}, x),
\tag{A-10}
$$

where can be deduced that the matrix also satisfies the same property $\bar{\boldsymbol{Q}}_t(x, x|\boldsymbol{c}) + \sum_{x' \neq x} \bar{\boldsymbol{Q}}_t(x', x|\boldsymbol{c}) = 0$. Therefore, we can rewrite Equation (A-9) as:

$$
\begin{aligned}
p(x_{t-\Delta t} = \hat{x}|x_t = x, \boldsymbol{c}) &= \delta_{\hat{x},x}\left(1 + \bar{\boldsymbol{Q}}_t(x, x|\boldsymbol{c})\Delta t\right) + (1 - \delta_{\hat{x},x})\bar{\boldsymbol{Q}}_t(\hat{x}, x|\boldsymbol{c})\Delta t + o(\Delta t) \\
&= \delta_{\hat{x},x} + \bar{\boldsymbol{Q}}_t(\hat{x}, x|\boldsymbol{c})\Delta t + o(\Delta t).
\end{aligned}
\tag{A-11}
$$

Furthermore, to achieve predictor-free guidance (Ho & Salimans, 2021; Nisonoff et al., 2024), we use the Bayes rule to relive the dependence on any predictor/classifier:

$$
\begin{aligned}
\bar{\boldsymbol{Q}}_t(\hat{x}, x|\boldsymbol{c}) &= \frac{p(\boldsymbol{c}|x_{t-\Delta t} = \hat{x}, x_t = x)}{p(\boldsymbol{c}|x_{t-\Delta t} = x, x_t = x)}\bar{\boldsymbol{Q}}_t(\hat{x}, x) \\
&= \frac{p(x_{t-\Delta t} = \hat{x}|x_t = x, \boldsymbol{c})\cancel{p(\boldsymbol{c}|x_t=x)}}{p(x_{t-\Delta t} = \hat{x}|x_t = x)} \frac{p(x_{t-\Delta t} = x|x_t = x)}{p(x_{t-\Delta t} = x|x_t = x, \boldsymbol{c})\cancel{p(\boldsymbol{c}|x_t=x)}}\bar{\boldsymbol{Q}}_t(\hat{x}, x) \\
&= \frac{p(x_{t-\Delta t} = \hat{x}|x_t = x, \boldsymbol{c})}{p(x_{t-\Delta t} = x|x_t = x, \boldsymbol{c})} \frac{p(x_{t-\Delta t} = x|x_t = x)}{p(x_{t-\Delta t} = \hat{x}|x_t = x)}\bar{\boldsymbol{Q}}_t(\hat{x}, x) \\
&\approx \frac{s_\theta(x, t, \boldsymbol{c})_{\hat{x}}}{s_\theta(x, t)_{\hat{x}}}\bar{\boldsymbol{Q}}_t(\hat{x}, x),
\end{aligned}
\tag{A-12}
$$

where we utilize the concrete score $s_\theta(x, t, \boldsymbol{c})_{\hat{x}}$ in Equation (2) to estimate the ratio like $\frac{p(x_{t-\Delta t}=\hat{x}|x_t=x,\boldsymbol{c})}{p(x_{t-\Delta t}=x|x_t=x,\boldsymbol{c})}$. Similar to previous methods, we can introduce a guidance scale $w$ (i.e. guidance strength) as:

$$
\bar{\boldsymbol{Q}}_t(\hat{x}, x|\boldsymbol{c}) \approx \frac{s_\theta^w(x, t, \boldsymbol{c})_{\hat{x}}}{s_\theta^w(x, t)_{\hat{x}}}\bar{\boldsymbol{Q}}_t(\hat{x}, x) = \frac{s_\theta^w(x, t, \boldsymbol{c})_{\hat{x}}}{s_\theta^w(x, t)_{\hat{x}}}\left(s_\theta(x, t)_{\hat{x}}\bar{\boldsymbol{Q}}_t^{\text{tok}}(\hat{x}, x)\right) = \frac{s_\theta^w(x, t, \boldsymbol{c})_{\hat{x}}}{s_\theta^{w-1}(x, t)_{\hat{x}}}\boldsymbol{Q}_t^{\text{tok}}(\hat{x}, x)
\tag{A-13}
$$

where $\boldsymbol{Q}^{\text{tok}}$ is the fixed diffusion transition rate matrix in Equation (A-3). Since $\boldsymbol{c} = \{\boldsymbol{c}_1, \ldots, \boldsymbol{c}_k\}$ contains $k$ independent conditions, we can rewrite Equation (A-12) into multi-conditional form as:

$$
\begin{aligned}
\bar{\boldsymbol{Q}}_t(\hat{x}, x|\boldsymbol{c}_1, \ldots, \boldsymbol{c}_k) &= \bar{\boldsymbol{Q}}_t(\hat{x}, x)\prod_{i=1}^k \frac{p(\boldsymbol{c}_i|x_{t-\Delta t} = \hat{x}, x_t = x)}{p(\boldsymbol{c}_i|x_{t-\Delta t} = x, x_t = x)} \\
&= \bar{\boldsymbol{Q}}_t(\hat{x}, x)\prod_{i=1}^k \frac{p(x_{t-\Delta t} = \hat{x}|x_t = x, \boldsymbol{c}_i)}{p(x_{t-\Delta t} = x|x_t = x, \boldsymbol{c}_i)} \frac{p(x_{t-\Delta t} = x|x_t = x)}{p(x_{t-\Delta t} = \hat{x}|x_t = x)} \\
&\approx \bar{\boldsymbol{Q}}_t(\hat{x}, x)\prod_{i=1}^k \frac{s_\theta^{w_i}(x, t, \boldsymbol{c}_i)_{\hat{x}}}{s_\theta^{w_i}(x, t)_{\hat{x}}} \\
&= \left[s_\theta(x, t)_{\hat{x}}\prod_{i=1}^k \frac{s_\theta^{w_i}(x, t, \boldsymbol{c}_i)_{\hat{x}}}{s_\theta^{w_i}(x, t)_{\hat{x}}}\right]\boldsymbol{Q}_t^{\text{tok}}(\hat{x}, x).
\end{aligned}
\tag{A-14}
$$

Energy-Based Models (EBMs) (Guo et al., 2023a; Geng et al., 2024; Liu et al., 2022) are a class of generative models and also known as non-normalized probabilistic models. Given speech token sequence $\boldsymbol{x}$ and a learnable neural network $f_\theta$, the probability distribution of EBM can be formulated as:

$$
p_\theta(\boldsymbol{x}) = \frac{e^{f_\theta(x)}}{Z},
\tag{A-15}
$$

where $Z = \sum_{\boldsymbol{x} \in \mathcal{X}} e^{f_\theta(x)}$ is a normalizing constant, and $f_\theta$ is the energy function. Inspired by the formulation of the EBM, the score can also be formulated as $\hat{s}_\theta(x)_{\hat{x}} \approx \frac{p_\theta(\hat{x})}{p_\theta(x)} = \frac{e^{f_\theta(\hat{x})}/Z}{e^{f_\theta(x)}/Z} = \frac{e^{f_\theta(\hat{x})}}{e^{f_\theta(x)}}$, where $x = x_t, \hat{x} = x_{t-\Delta t}$, $Z$ is the normalizing constant, and $f_\theta$ is the energy function. As we typically define the energy function as a sum of multiple terms (Kim & Bengio, 2016), we can associate each term with the joint and compositional ones, and the final probability distribution is expressed as a product of both. Hence, we can obtain the modulated score $\hat{s}_\theta(x, t)_{\hat{x}}$ by multiplying the compositional score and joint score (i.e. sum up the energy functions):

$$
\hat{s}_\theta^{(w)}(x, t)_{\hat{x}} = \underbrace{s_\theta(x, t)_{\hat{x}}\prod_{k=1}^K \frac{s_\theta^{w_i}(x, t, \boldsymbol{c}_k)_{\hat{x}}}{s_\theta^{w_i}(x, t)_{\hat{x}}}}_{\text{Compositional}} \cdot \underbrace{\frac{s_\theta^{w_0}(x, t, \boldsymbol{c})_{\hat{x}}}{s_\theta^{w_0-1}(x, t)_{\hat{x}}}}_{\text{Joint}},
\tag{A-16}
$$

where $\boldsymbol{c} = \{\boldsymbol{c}_{\text{id}}, \boldsymbol{c}_{\text{emo}}, \boldsymbol{c}_{\text{text}}\}$, $w_0$ controls the scale of guidance strength for the joint injection of all conditions, while $w_i$ for $1 \leq i \leq k$ is assigned to each independent attribute (*i.e.* identity, emotion, and semantics with $k = 3$). Therefore, we can rewrite Equation (A-11) as:

$$
\begin{aligned}
p(x_{t-\Delta t} = \hat{x} | x_t = x, \boldsymbol{c}) &= \delta_{\hat{x},x} + \bar{\boldsymbol{Q}}_t(\hat{x}, x | \boldsymbol{c}) \Delta t + o(\Delta t) \\
&\approx \delta_{\hat{x},x} + \hat{s}_\theta^{(w)}(x,t)_{\hat{x}} \, \boldsymbol{Q}_t^{\text{tok}}(\hat{x}, x) \Delta t + o(\Delta t)
\end{aligned}
\tag{A-17}
$$

## D. Datasets

### D.1. Dataset Statistics

All our models are pre-trained on three datasets with pairs of face video and speech: RAVDESS (Livingstone & Russo, 2018), MEAD (Wang et al., 2020; Gan et al., 2023), and MELD-FAIR (Carneiro et al., 2023). The RAVDESS contains 1,440 English utterances voiced by 12 male and 12 female actors with eight different emotions. The MEAD is a talking-face video corpus featuring 60 actors and actresses talking with eight different emotions at three different intensity levels. The MELD-FAIR introduces a novel pre-processing pipeline to fix noisy alignment issues of the MEAD (Wang et al., 2020) consisting of text-audio-video pairs extracted from the *Friends* TV series. Then, for the training, we train all our models using a combination of all three datasets. The RAVDESS and MEAD of the combined one are randomly segmented into training, validation, and test sets without any speaker overlap. In contrast, we follow the original splits of the MELD-FAIR dataset with speaker overlap. Additionally, these datasets lack sufficient semantic units in real-world environments, making it challenging to train a TTS model. We incorporate a 10-hour subset from LRS3 (Afouras et al., 2018) for pre-training, allowing the model to be comparable to Face-TTS trained on 400 hours of LRS3. Finally, the combined dataset comprises 31.33 hours of audio recordings and 26,767 utterances across 7 basic emotions (*i.e.* angry, disgust, fear, happy, neutral, sad, and surprised) and 953 speakers.

### D.2. Data Preprocessing Details

For data pre-processing, considering the presence of non-primary speakers and background noise such as audience interactions in the recordings, we first resample the audio to a single-channel 16-bit at 16 kHz format, then apply Sep-Former (Subakan et al., 2021), a state-of-the-art model in speech separation, to isolate the primary speaker's audio and reduce noise from other voices. Then, we introduce an automatic speech recognition model Whisper (Radford et al., 2023) to filter non-aligned text-speech pairs (*i.e.* WER higher than 10%).

## E. Model Details

### E.1. Implementation Details of DEmoFace

Table A-2 shows more details about our DEmoFace. Firstly, for our multimodal diffusion transformer, it contains 12 MM-DiT blocks, with channel numbers 768, attention heads 12 for each block. We train the model using the AdamW optimizer (Loshchilov & Hutter, 2019) with $\beta_1 = 0.9$, $\beta_2 = 0.999$, a learning rate of 1e-4, batch size 32, and a 24GB NVIDIA RTX 4090 GPU. The total number of iterations is 300k. For a fair comparison, we do not perform any pre-training or fine-tuning on the test set. During inference, we use the Euler sampler with 96 steps following (Lou et al., 2024).

Secondly, we train our identity encoder achiving face-speech alignment on a 24GB NVIDIA 4090 GPU, with a total batch size of 12 samples. We use the AdamW optimizer (Loshchilov & Hutter, 2019) with $\beta_1 = 0.9$, $\beta_2 = 0.999$, $\epsilon = 10e-9$. It takes 80k steps for training until convergence.

Lastly, we design frame-level duration predictor to predict the target speech duration during inference, which obtains the total duration of the target speech through summing up the phoneme-level inputs. We directly estimate the total target speech duration instead of the phoneme-level durations. The duration predictor has three convolution layers and a MLP architecture to predict duration from the frozen SpeechT5 (Ao et al., 2022) encoder. The predictor is trained using the AdamW optimizer with 0.9 and 0.999. The initial learning rate is set to 1e-4 with a learning rate decay of 0.999. We use a total batch size of 32 and train the model with 1 NVIDIA 4090 GPUs at least 100k steps.

| Model | Configuration | Parameter |
|---|---|---|
| Multimodal Diffusion Transformer | In / Out Channels | 1 / 1 |
| | Number of Transformer Blocks | 12 |
| | Hidden Channel | 768 |
| | Attention Heads | 12 |
| | $c^{\text{id}}$ Identity Embedding Dimension | 256 |
| | $c^{\text{emo}}$ Emotion Embedding Dimension | 128 |
| | $c^{\text{text}}$ Text Embedding Dimension | 768 |
| | Activate Function | SiLU |
| | Dropout | 0.1 |
| Speech Codec | Input | Waveform |
| | Sampling Rate | 24kHz |
| | Hopsize | 480 |
| | Number of RVQ Blocks | 12 |
| | Codebook size | 1024 |
| | Coodbook Dimension | 8 |
| | Decoder Hidden Dimension | 512 |
| | Decoder Kernel Size | 12 |
| | Number of Decoder Blocks | 30 |
| Identity Encoder | ArcFace-Net Output Dimension | 512 |
| | FaceNet Output Dimension | 512 |
| | MLP Channels | (512, 512, 256, 256) |
| | Activate Function | GeLU |
| Duration Predictor | Input | SpeechT5 Text Embedding |
| | Conv Channel | 256 |
| | Conv Kernel | 5 |
| | MLP Channels | (256, 1) |
| | Activate Function | ReLU |

Table A-2: Implementation details about our DEmoFace.

## E.2. Implementation Details of Baselines

**EmoSpeech.** Accroding to EmoSpeech (Diatlova & Shutov, 2023) official code[1], we reproduce training process on our pre-training dataset. EmoSpeech introduces a conditioning mechanism that captures the relationship between speech intonation and the emotional intensity assigned to each token in the sequence. Then we train EmoSpeech following the original setting of its paper. We use the Adam optimizer (Kingma & Ba, 2015) with $\beta_1 = 0.5$, $\beta_2 = 0.9$, $\epsilon = 10^{-9}$ and follow the same learning rate schedule in vanilla transformer (Vaswani et al., 2017). It takes 300k steps with batch size 64 for training until convergence in a single GPU. In the inference process, the output mel-spectrograms of the EmoSPeech are also transformed into speech samples using the pre-trained vocoder[2].

**FastSpeech 2.** Since FastSpeech 2 (Ren et al., 2021) is not open source and emotion-awareness, we reproduce its method on our pre-training dataset based on the code[3] and its emotion-aware version on V2C-Net (Cong et al., 2023). To model the emotion-awareness in FastSpeech 2, following previous methods (Chen et al., 2022; Cong et al., 2023), we utilize emotion embeddings from an emotion encoder I3D (Carreira & Zisserman, 2017) and speaker embeddings extracted via a generalized end-to-end speaker verification model (Wan et al., 2018) as additional inputs. These embeddings are projected and added to hidden embeddings before the variance adaptor. Then we train FastSpeech 2 following the original setting of its paper. We use the Adam optimizer (Kingma & Ba, 2015) with $\beta_1 = 0.9$, $\beta_2 = 0.98$, $\epsilon = 10^{-9}$ and follow the same

---

[1]https://github.com/deepvk/emospeech
[2]https://github.com/jik876/hifi-gan
[3]https://github.com/ming024/FastSpeech2

learning rate schedule in vanilla transformer (Vaswani et al., 2017). It takes 300k steps with batch size 48 for training until convergence. In the inference process, the output mel-spectrograms of the FastSpeech 2 are also transformed into speech samples using the pre-trained vocoder.

**V2C-Net.** The V2C-Net (Chen et al., 2022) is not open source, so we reproduce its method based on its original paper and project[4]. To exploit the emotion from the reference video, it utilizes an emotion encoder I3D (Carreira & Zisserman, 2017) to calculate the emotion embedding and proposes a speaker encoder comprising 3 LSTM layers and a linear layer to explore the voice characteristics of different speakers. Then we train V2C-Net on our pre-training dataset according to the setup outlined in the original paper. The Adam optimizer (Kingma & Ba, 2015) is employed with hyperparameters set to $\beta_1 = 0.9$, and $\beta_2 = 0.98$. The learning rate schedule followed the approach used in the vanilla transformer (Vaswani et al., 2017). It takes 300k steps with a batch size of 48. During inference, the generated mel-spectrogram is converted into speech using the pre-trained vocoder.

**HPM.** According to HPM (Cong et al., 2023) official code[5], we reproduce training process on our pre-training dataset. It utilizes an emotion face-alignment network (EmoFAN) (**?**) to capture the valence and arousal information from facial expressions and also utilizes an emotion encoder I3D (Carreira & Zisserman, 2017) to calculate the emotion embedding. For training, we use Adam (Kingma & Ba, 2015) with learning rate $10^{-5}$, $\beta_1 = 0.9$, $\beta_2 = 0.98$, $\epsilon = 10^{-9}$ to optimize the HPM. It takes 500k steps with batch size 16. During inference, the generated mel-spectrogram is converted into speech using the pre-trained vocoder.

**StyleDubber.** According to StyleDubber (Cong et al., 2024b) official code[6], we reproduce training process on our dataset CMC-TED. StyleDubber introduces the cross-attention to enhance the relevance between textual phonemes of the script and the reference audio as well as visual emotion. For training, we use Adam (Kingma & Ba, 2015) with learning rate 0.00625, $\beta_1 = 0.9$, $\beta_2 = 0.98$, $\epsilon = 10^{-9}$ to optimize the model. It takes 300k steps with batch size 64. During inference, the generated mel-spectrogram is converted into speech using the pre-trained vocoder.

**Face-TTS.** We use the official-released pre-trained model[7] of the Face-TTS (Lee et al., 2023), which is pre-trained on multiple large-scale TTS datasets (such as LRS3 (Afouras et al., 2018), VoxCeleb2 (Chung et al., 2018), and LJSpeech (Ito & Johnson, 2017), etc.). Following its original inference pipeline, the input face image is resized into 224×224 pixels and embeds onto 512-dimensional vector. The output speech is decoded from their released vocoder in 16kHz sampling rate.

## F. Additional Results

We conduct extra experiments under our acoustic-guided version DEmoFace *, as shown in Fig. A-1, from mel-spectrograms in the second row, the other baselines show severe over-smoothing issues, resulting quality degradation. Furthermore, from the F0 curve in the second row, the other baselines exhibit distinct F0 contours showing different pitch, emotion, and intonation with the GT. Our results are closer to the GT with those acoustic-guided methods.

*For More audio samples please refer to our supplementary material.*

---

[4]https://github.com/chenqi008/V2C
[5]https://github.com/GalaxyCong/HPMDubbing
[6]https://github.com/GalaxyCong/StyleDubber
[7]https://github.com/naver-ai/facetts

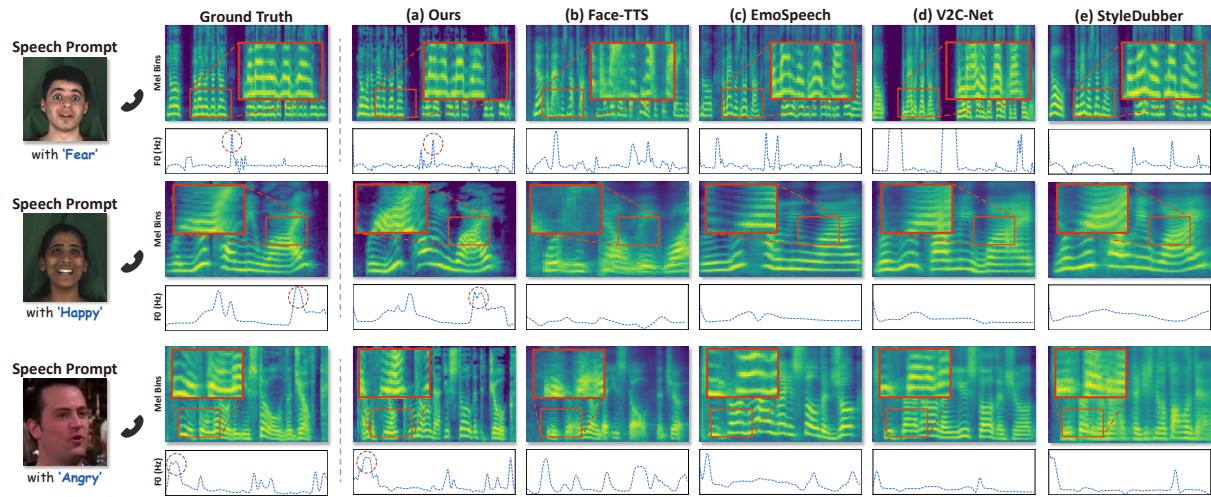

Figure A-1: **Speech qualitative results** on acoustic-guided version DEmoFace *. The red rectangles highlight key regions with acoustic differences or over-smoothing issues, and the red dotted circle shows similar F0 contours with ground truth.

## G. User Evaluation

We further conduct the subjective evaluation with 50 participants, to compare our DEmoFace with SOTA methods (Diatlova & Shutov, 2023; Lee et al., 2023). Specifically, we introduce three metrics based on five mean opinion scores (MOS), each rated on a scale from 1 to 5 in 0.5 increments. Among them, $MOS_{nat}$ evaluates speech naturalness and generation quality, while $MOS_{id}$, and $MOS_{emo}$ assess the perceived similarity of timbre and emotional prosody, respectively. We randomly generate 15 utterances from the test set for subjective evaluation.

We conduct the subjective evaluation with 15 participants, to compare our DEmoFace with SOTA methods. Specifically, we introduce five mean opinion scores (MOS) with rating scores from 1 to 5 in 0.5 increments, including $MOS_{nat}$, $MOS_{con}$ for speech naturalness (*i.e.* quality) and consistency (*i.e.* emotion and speaker similarity). We randomly generate 10 samples from the test set. Here, we give definitions of both MOS scores on Tables A-3.

| Level | Value | $MOS_{nat}$ *Description* | $MOS_{id}$ *Description* | $MOS_{emo}$ *Description* |
|---|---|---|---|---|
| **Excellent** | 5 | Natural, fluent, and clear speech with no rhythm or semantic issues. High intelligibility and quality. | Timbre strongly reflects the speaker identity as inferred from the facial cues, including gender and age. | Emotion expression closely matches the expression cues in the image. |
| **Good** | 4 | Mostly clear speech with minor rhythm issues. Acceptable overall quality. | Timbre generally aligns with the imagined identity, with slight deviations in age or gender cues. | Emotion is largely consistent with the facial expression, with slight differences. |
| **Fair** | 3 | Noticeable unnaturalness and errors; speech requires effort to understand. Moderate quality. | Timbre partially corresponds to the visual identity cues but with evident inconsistencies. | Emotion partially matches facial cues but with evident inconsistencies. |
| **Bad** | 2 | Disfluent speech with abnormal rhythm and unclear articulation. Low intelligibility and quality. | Timbre poorly aligns with the inferred identity, showing significant mismatch. | Emotion poorly aligned with facial expression. |
| **Poor** | 1 | Unintelligible, highly disfluent, and unclear speech. Unacceptable quality. | Timbre shows no meaningful relation to the imagined speaker identity. | Emotion completely inconsistent with visual cues. |

Table A-3: $MOS_{nat}$, $MOS_{id}$, and $MOS_{emo}$ descriptions.

