# OpenReview forum: "Emotional Face-to-Speech"
_ICML.cc/2025/Conference — ICML 2025 poster_

### Official Review · Reviewer_4KNc · 2025-03-11

**Overall Recommendation:** 3

**Summary:**

This paper describes an approach for mapping silent video of a talking face to the synthesized voice.  The approach is based on a discrete diffusion transformer that is conditioned on the (visual) speaker identity and a learned representation of the facial expression of emotion.  Together these help to preserve speaker identity and improve the expressiveness of the speech.   In addition, residual vector quantization is used learn a coarse-to-fine tokenization to better capture voice characteristics at different levels of granularity.  The approach is evaluated using both objective and subjective assessment against a number of baselines.

### Update
I will maintain my score and rate this paper weak accept. The approach used makes sense, but I am disappointed in the evaluation in that the baselines the authors are comparing against are not strictly the true baselines.  The authors of the original baselines used different datasets, and the authors here can claim their approach beats the baselines on their experimental configuration, but the "standard" baselines do much better than the examples presented here.

**Claims And Evidence:**

I believe the claims are backed by the experiments in the paper.

**Essential References Not Discussed:**

A potentially relevant reference that is missing:  Lu, J.; Sisman, B.; Liu, R.; Zhang, M.; and Li, H. . VisualTTS: TTS with Accurate Lip-Speech Synchronization for Automatic Voice Over. In Proceedings of ICASSP.  2022.

Most of the baselines (e.g., from Table 1) are not discussed in the related work section.  It would be useful to have the work situated within that broader literature.  Explain what their individual limitations are, etc.

**Experimental Designs Or Analyses:**

I do not have specific concerns about the experimental design.  It is good to see a combination of objective and subjective assessment of the approach.  I did wonder about asking viewers about rating the identity and expressiveness separately.  How much does each individual attribute contribute to the goodness/degradation in the perceived consistency?

**Methods And Evaluation Criteria:**

The datasets used for training and evaluating are standard.  Reasonable baselines have been used to benchmark against too, including face-driven methods and speech-driven methods.

**Other Comments Or Suggestions:**

There are places where the word choice or use of incomplete sentences are use makes the paper difficult to follow without having to re-reading.  For example, the first sentence of the description of the forward diffusion process (line 141) is not a sentence, the sentence beginning at line 152 is poorly written,  and the last sentence before the description of the training objective in Section 3 is also not a sentence.

Expand the caption for Figure 4 to more completely describe the figure.

In Figure 5, the relative training cost is provided.  Does this mean that “baseline” and “ours” are using a different number of steps, or are they using the same number of steps within a sub-figure, but a different number between the WER, emotion/speaker similarity figures?

In the introduction you mention: “Considering that facial expressions are the most direct indicators of emotion“ — it might be worth qualifying this and saying facial expressions are the most VISUAL indictor — tone of voice and other acoustic cues are equally indicative of emotion state too.

In the Datasets section of Section 5, what does “Additionally, these datasets lack sufficient semantic units in real-world environments, making it challenging to train a TTS model.” mean?  What are the “sufficient semantic units”?  Do you mean these audiovisual datasets are too small in terms of sample size (hours of speech) to train a high-quality TTS system?

**Other Strengths And Weaknesses:**

Strengths:
+ The use of RVQ/curriculum learning and conditioning on the representation of the facial expression clearly improve the quality of the generated speech.
+ The approach is significantly better then the baseline as evidenced by the results of the objective and subjective tests, and listening to the provided samples (albeit with the concern I have about the provided samples from the baselines that I highlight elsewhere),.

Weaknesses:
- It would have been nice to have more examples to see how other factors affect the approach.  For example, varying the degree of expressiveness from mid-expressive to very expressive.  How does this affect generation quality, and does the relative degree affect the contribution of the identity/expression conditioning on the network?

**Questions For Authors:**

Q1:  In the introduction, the wording refers to “... the one-to-many issues inherent in continuous speech features." — what are these one-to-many issues?  Given the context of the work (lip motion to speech mapping) I was wondering if it is the mapping of a visual lip gesture, e.g., lip closure of a bilabial plosive, mapping to many speech sounds, e.g., /b/, /p/, /m/.  This does not sit with "continuous speech features." though so I am unclear.

Q2:  For Figure 4 — these are all data points from synthetic speech.  How do data points for synthetic vs. real speech align in this low-dimensional projection?  For example, are all of the data points for the synthetic voice of a speaker aligned with/distinct from the data points for the corresponding real voice.  What is presented here shows that the performance of DemoFace aligns better than the baseline, but that is only part of the story.

Q3:  I have a concern over the examples used in the baselines that are provided by way of comparison.  The voice quality in some of the baselines is considerably worse than the quality suggested on the original demo page(s).  Why is this the case?  From what I understand you are using the original implementations provided by the authors, and in many cases the same datasets (which suggests it is not a distribution shift that might require different hyper-parameters).,

**Relation To Broader Scientific Literature:**

There are two main contributions here:  1) conditioning the discrete diffusion transformer on the facial expression representation in addition to the identity representation, and 2) using residual vector quantization, which has been shown to be important in learning speaker representations.

**Theoretical Claims:**

No, I did not check the derivations in the appendix beyond skimming them for information.

---

> ### Author Rebuttal · Authors · 2025-04-01
>
> We are grateful for your kind words and appreciating the significance of our contributions, and we try our best to address your questions as follows.
>
>
> **Q1: Impact of expressiveness variation**
>
> Thank you for the insightful suggestion. In this paper, we use one-hot emotion labels to learn identity-agnostic emotional embeddings, ensuring that variations in expression do not significantly affect the generation results. In the future, we plan to incorporate emotional intensity as an additional condition to achieve more natural speech synthesis.
>
>
>
> **Q2: Extra user study**
>
> Thank you for the valuable suggestion. Per your suggestion, we have supplemented an extra objective evaluation with larger participants and new MOS metrics for both attributes. Please refer to the Q5 response for Reviewer 7YBQ for the results, which show that DEmoFace outperforms in naturalness, identity timbre, and emotional prosody consistency. We will revise our manuscript accordingly.
>
>
>
> **Q3: Clarification on one-to-many mapping issue in continuous speech features**
>
> The one-to-many mapping of speech generation denotes that multiple speech sequences can possibly correspond to the same text sequence with diverse pitch, duration and prosody, making the synthesis speech distribution multimodal instead of unimodal. The issue arises from attempting to generate continuous speech features, such as mel-spectrograms, which are highly correlated over time and frequency, leading to over-smoothing during frame-level prosody or linguistic predictions [1]. In addition, in this paper, we focus on extracting identity styles and emotions from facial features, rather than semantics from lip motion. We will certainly clarify this in future version.
>
> [1]: FoundationTTS: Text-to-Speech for ASR Customization with Generative Language Model
>
>
>
> **Q4: Clarification on t-SNE experiments**
>
> To further compare our method with real speech data and ablation methods, we have added relevant visualizations referring to Figures 1 and 2 in the anonymous link https://anonymous.4open.science/r/demoface. The results demonstrate that our method shows a similar clustering distribution to that of real speech.
>
>
>
> **Q5: Implementation of baselines**
>
> To ensure a fair comparison, we have included detailed implementation details in the appendix. The speech quality differences in some baselines, compared to original demos, stem from their limited dataset size and lenient evaluation, which may lead to overfitting. For example, they relied on the limited-vocabulary GRID dataset or the speaker-limited V2C dataset. In contrast, our method introduces a mixed dataset and is evaluated on unseen speakers,  reflecting more challenging real-world scenarios.
>
>
>
> **Q6: Clarification on related works, references, and writing**
>
> We appreciate your valuable feedback, which has helped us refine our work.
>
> 1. We will revise the related work section in detail: Compared with acoustic-guided methods, these non-autoregressive (NAR) methods suffer from over-smoothing, less diversity, and complex alignment issues, despite fast inference. Compared with visual-guided methods, they still pose the above issues by introducing NAR for coarse mel-spectrogram generation and diffusion for refinement and lacking an efficient conditioning paradigm.  In contrast, our DEmoFace can flexibly leverage both visual and acoustic conditions while dynamically aligning text for higher-quality and more diverse speech generation, offering practical guidance for optimizing the promising modeling paradigm.
> 2. We will add the suggested references and compare them with ours.
> 3. We will refine our writing to improve clarity and enhance readability.
>
>
>
> **Q7: Training cost comparison in Figure 5**
>
> Figure 5 of main text shows a comparison of metrics in the same model checkpoint, under the same training steps or costs, which demonstrates that curriculum learning can improve training efficiency.
>
>
>
> **Q8: Should facial expressions be called the ``most visual indicator''**
>
> Our proposed eF2S problem aims to infer timbre and emotional prosody solely from visual cues instead of acoustic cues, with facial expressions serving as the most relevant source of emotional information naturally.
>
>
>
> **Q9: Why need extra an LRS3 subset for learning more semantic units**
>
> The "semantic unit" refers to text-rich units containing contextualized linguistic details. The audiovisual datasets used in this work exhibit limited linguistic diversity in acted emotional speech, which constrains the model to accurately generate speech for unseen text. Therefore, we incorporate a subset of LRS3 to enhance semantic learning from real-world scenario.

---

> > ### Comment · Reviewer_4KNc · 2025-04-03
> >
> > Thank you for the detailed responses.
> >
> > For Q5:  I am still a little unsure why, if you are using the code provided by the authors of the baselines and have used the same data, the example outputs from their system that you have created sound worse (sometimes significantly so) than the examples on the original demo pages. I appreciate what you are saying about limited data, e.g., GRID, being problematic, but if the original authors used the same data would one not expect the equivalent quality when you recreate samples?
> >
> > For Q9:  are you referring to "limited phonetic coverage"? This is what the phenomenon used to be referred to in the speech community when a model has not seen/heard phonemes in a sufficient number of contexts, so the model cannot produce the sound with appropriate coarticulation effects taken into account.

---

> > > ### Author Response · Authors · 2025-04-03
> > >
> > > Thank you for your additional questions. We sincerely apologize for any lack of clarity in our previous responses, partly due to space constraints. We appreciate this extra opportunity to address your remaining concerns as thoroughly as possible.
> > >
> > > **Responses to Q5**: The performance decline of baselines can be attributed to two key factors.
> > >
> > > 1. **Differences in Training Data:** We fully agree with you that baselines trained on the same data with the same code should yield consistent results. However, this study introduces a new task and a corresponding dataset, which differs from those used in the original baselines. To ensure a fair comparison, we re-trained the baselines using the same training dataset as the proposed DEmoFace while strictly adhering to their original configurations. Here, we would like to highlight that the change of training data can significantly impact generation performance. For example, as shown on the StyleDubber demo page (https://acl2024x.github.io/StyleDubber/#Setting2), one can observe that models trained on the smaller Grid dataset outperform those trained on the larger V2C dataset in generation quality. A possible reason is that the distribution shift between training and test sets in Grid dataset is smaller than that in V2C dataset. Additionally, in our response to Q2 for Reviewer kkEx, our dataset-wise experiments further demonstrate significant performance variations of the same model across different training datasets. Therefore, the performance of these baselines on our more diverse and realistic dataset may differ from the originally reported results.
> > > 2. **Differences in Experimental Setup:** In this study, we focus on evaluating the generalization ability of all methods in real-world scenarios by ensuring no speaker overlap between the training and testing sets, which enforces a stricter constraint and hence, makes speech generation more challenging. Here, we would like to highlight that the performance degradation across distinct setups can also be observed on the StyleDubber demo page (https://acl2024x.github.io/StyleDubber/). In the Dub 1.0 and 2.0 settings where the driven speech comes from speakers seen during training, all methods produced high-quality speech due to speaker information leakage, while limiting their applicability to real-world scenarios. However, in the Dub 3.0 setting where the driven speech comes from unseen speakers (**aligning with our setup**), the generated speech quality of all methods significantly declines due to insufficient generalization, exhibiting unclear pronunciation and audio distortion. Therefore, these differences in experimental setups could explain the performance degradation observed in baselines when tested on unseen speakers from a more diverse and realistic dataset in this study.
> > >
> > >
> > > We believe these two factors outlined above are the primary reasons for the observed discrepancies in generation quality. To facilitate the community in fully reproducing our results, we will open-source all the DEmoFace code as well as the re-training code for other baselines. We hope this will further address your concerns.
> > >
> > >
> > >
> > > **Responses to Q9**: Thank you for requesting clarification on limited phonetic coverage. Yes, we are referring to “limited phonetic coverage”, which occurs when a model struggles to generate accurate speech due to being trained on a dataset with a restricted set of phonemes. We appreciate your feedback and will update our manuscript to further clarify this concept.

---

### Official Review · Reviewer_7YBQ · 2025-03-13

**Overall Recommendation:** 3

**Summary:**

This paper argues that extracting and applying emotional expressions as well as identities when generating speech based on face prompt input is effective in resolving face-speech mismatch.

To this end, we propose an Emotional Face-to-Speech (eF2S) method that goes beyond the existing Face-to-Speech (F2S) method and applies emotions extracted from faces to the generated speech.
The proposed framework for eF2S (DEmoFace) generates speech by directly integrating both identity and emotional expressions from face input.

**Claims And Evidence:**

I agree that generating speech synchronized with emotion is essential for natural speech synthesis.
However, is it necessary to extract emotion from a single video frame?
For generating speech that aligns with emotional intent, it is reasonable to assume that text-based emotion should have a more significant influence than facial emotion.

The Emotion of mismatch between face and text can lead to inconsistencies in speech generation.
For example, if the image shows a smiling face, but the text expresses anger (e.g., “I am angry”), this would represent a perceptual conflicts between the generated speech expression and the text expression. In this paper does not provide a detailed discussion on how to handle such conflicts—specifically, which modality (face or text) should take precedence in determining the speech’s emotion.

**Essential References Not Discussed:**

In this paper, the generated speech is incorporated a emotion of face by extracting emotional features.

However, it lacks a discussion on how Facial Emotion Recognition has evolved and its limitations in speech emotion modeling.

**Experimental Designs Or Analyses:**

1. Lack of experiments about emotion condition:
The paper does not experimentally verify which factor—text emotion or face emotion—plays a more dominant role in generating speech emotion when they conflict.

2. Inconsistency in the interpretation of Figure 4 (t-SNE visualization):
The paper claims that emotion regulation leads to more natural speech synthesis, but Figure 4 (t-SNE visualization) primarily shows a distribution based on gender differences.

3. Lack of verification of face emotion to speech synthesis:
There is no evaluation of performance when generating speech using only identity $c_{id}$ and text $c_{text}$, without the emotion condition $c_{emo}$.

**Methods And Evaluation Criteria:**

In this paper, the authors did not provide a comparative evaluation across the datasets used in training. While it employs multiple datasets (RAVDESS, MEAD, MELD-FAIR, and LRS3), there is no analysis on how the model performs differently across these datasets.

Specifically, it is unclear whether emotion-rich datasets like RAVDESS lead to better emotion modeling compared to conversational datasets like MELD-FAIR. Additionally, there is no discussion on whether training on one dataset generalizes well to others.

**Other Comments Or Suggestions:**

In the introduction section, when explaining the limitations of existing methods compared to the authors’ approach, the explanation is difficult to follow. (L26~45)

**Other Strengths And Weaknesses:**

- Strengths:

In this paper is proposed a novel speech generation framework, incorporating emotion conditioning from facial features, which has been insufficiently explored in previous works.

- Weaknesses:

The lack of dataset-specific experiments makes it difficult to evaluate whether this model generalizes well.

The t-SNE visualization (Fig. 4) only shows gender-based clustering, which does not directly validate the effectiveness of emotional conditioning.

There is no ablation study for removing the emotion condition, which is critical to justify the importance of facial emotion in speech generation.

**Questions For Authors:**

1. How does the model resolve conflicts between text emotion and face emotion when they are contradictory?

2. Why does not measure dataset-wise comparison performance?

3. Why does Figure 4 focus on gender-based clustering instead of emotional clustering?

4. What would be the impact of removing the facial emotion condition in ablation studies?

**Relation To Broader Scientific Literature:**

This paper builds on prior work in Face-to-Speech (F2S) and expressive TTS by focusing on visual emotion conditioning, which has been underexplored in previous works.

Unlike previous TTS that primarily focused on identity-based speech generation, this work explicitly decouples identity and emotion to synthesize the speech that consistent emotion between speech and face.

**Theoretical Claims:**

Appendix B outlines the preliminaries of the discrete diffusion model, offering relevant definitions and concepts.

Appendix C presents the full derivation of the Enhanced Predictor-Free Guidance (EPFG) equations (Equations 4-5).

However, the explicit mathematical derivations for Equations (1-3) are not detailed in the supplementary materials.

---

> ### Author Rebuttal · Authors · 2025-04-01
>
> Thank you very much for your insightful comments and suggestions, motivating us to rethink a more comprehensive experiment evaluation. We try our best to address your questions as follows.
>
>
>
> **Q1: Emotion ambiguity between text and face**
>
> Thank you for your insightful comment. Determining the dominant and subordinate modalities is crucial for resolving ambiguities in multimodal emotion generation. However, in our eF2S task, we focus on inferring timbre and emotional prosody solely from visual cues, with facial expressions serving as the dominant modality rather than textual cues. In the future, we will explore integrating face, text, and speech for more consistent emotion generation by dynamically adjusting modality impacts through cross-modal shift estimation.
>
>
>
> **Q2: Dataset-wise experiment for out-of-domain robustness**
>
> We appreciate your insight for a more thorough evaluation of our method. To address your concern, we have supplemented dataset-wise experiments on the out-of-domain LRS2 benchmark. Specifically, due to the limited dataset size, we conducted full fine-tuning of the base model for 20 epochs on each of these four datasets to learn dataset-specific knowledge and tested them on LRS2. Please refer to the Q2 response for Reviewer kkEx for quantitative results, and we will include this evaluation and discussion in the future version.
>
>
>
> **Q3: Clarification on t-SNE experiment**
>
> To further compare our method with real speech data and ablation methods, we have added the requested visualizations, which can be found in Figures 1 and 2 in the anonymous link https://anonymous.4open.science/r/demoface. The results demonstrate that our method exhibits a clustering distribution similar to that of real speech.
>
>
>
> **Q4: Extra ablation study on emotion conditioning**
>
> Thank you for helping improve the clarity of our paper. We have conducted an ablation study on the $\boldsymbol{c} _\text{emo}$  condition. The results below show that $\boldsymbol{c} _\text{emo}$   enhances speech naturalness and expressiveness without significantly affecting SpkSim or WER, confirming that DEmoFace effectively decouples different conditions. The associated t-SNE visualizations can be found in the anonymous link provided above.
>
> | Methods                        | EmoSim$\uparrow$ | SpkSim$\uparrow$ | RMSE$\downarrow$ | MCD$\downarrow$ | WER$\downarrow$ |
> | ------------------------------ | ---------------- | ---------------- | ---------------- | --------------- | --------------- |
> | wo $\boldsymbol{c} _\text{emo}$ | 0.64             | 0.65             | 104.92           | 7.29            | 21.35           |
> | **DEmoFace**                   | **0.70**         | **0.67**         | **101.18**       | **6.86**        | **20.78**       |
>
>
>
> **Q5: Limited subjective evaluation**
>
> Following the setup of Face-TTS [1] using 17 evaluators, we initially conducted similar evaluations, and we acknowledge that a larger number of evaluators ensures more reliable results. Per your suggestions, we have expanded the evaluation to 50 evaluators with 15 samples each, and we also introduce new MOS metrics for both timbre （MOS$ _\text{id}$） and prosody （MOS$ _\text{emo}$） evaluation as Reviewer 4KNc suggested. The new evaluation results with 95% confidence intervals are as follows:
>
> | Methods   | MOS$ _\text{nat}\uparrow$ | MOS$ _\text{id}\uparrow$ | MOS$ _\text{emo}\uparrow$ |
> | --------- | ------------------------ | ----------------------- | ------------------------ |
> | EmoSpeech | 2.30±0.19                | 2.93±0.09               | 2.78±0.13                |
> | Face-TTS  | 2.28±0.09                | 2.67±0.12               | 2.75±0.09                |
> | **DEmoFace**  | **3.17**±0.18                | **3.20**±0.11               | **3.26**±0.12                |
>
> The results show that DEmoFace outperforms EmoSpeech and Face-TTS in naturalness, identity timbre, and emotional prosody consistency. We will revise our manuscript accordingly.
>
> \[1\] Imaginary Voice: Face-styled Diffusion Model for Text-to-Speech
>
>
>
> **Q6: Clarification on existing method limitations**
>
> The limitations of existing methods can be categorized into two aspects: 1) **task-level**, where they cannot jointly model speaker identity and emotion solely from visual cues, and 2) **method-level**, which involves issues like one-to-many mapping with limited diversity or inefficiencies in continuous NAR or discrete AR frameworks. Building on these insights, we propose a novel discrete diffusion framework for a novel task—emotional face-to-speech generation.
>
>
>
> **Q7: References to be discussed**
>
> Thank you for your suggestion. We will certainly incorporate the relevant discussion and citations [2-3] in the future version.
>
> \[2\]: Deep facial expression recognition: A survey \
> \[3\]: Emotion Recognition and Generation: A Comprehensive Review of Face, Speech, and Text Modalities

---

### Official Review · Reviewer_kkEx · 2025-03-13

**Overall Recommendation:** 3

**Summary:**

The paper introduces a task named Emotional Face-to-Speech (eF2S), which aims to synthesize emotional speech directly from expressive facial cues. The proposed DEmoFace leverages a discrete diffusion transformer with curriculum learning to achieve the SOTA eF2S performance.

**Claims And Evidence:**

The claims made in the paper are generally well-supported by clear and convincing evidence. The authors provide extensive experimental results, including both quantitative and qualitative evaluations.

**Essential References Not Discussed:**

None

**Experimental Designs Or Analyses:**

I have checked the experimental designs. It seems the authors have provided comprehensive experiments in the main text and appendix.

**Methods And Evaluation Criteria:**

The proposed methods and evaluation criteria are well-suited for the task. One potential limitation is the dataset size and diversity. While the authors use a combination of RAVDESS, MEAD, and MELD-FAIR, these datasets are relatively small and may not fully capture the variability in real-world scenarios.

**Other Comments Or Suggestions:**

None

**Other Strengths And Weaknesses:**

Weaknesses:
1. Limited Novelty and Efficiency:
The paper builds on existing techniques (e.g., discrete diffusion models, curriculum learning) without introducing fundamentally new algorithms. While the combination of these methods is creative, the lack of novel theoretical or algorithmic contributions limits the paper's originality. Additionally, the authors do not discuss the efficiency of the proposed framework, such as computational cost or inference speed, which is critical for real-world applications.
2. Limited Dataset Size and Diversity:
The experiments are conducted on relatively small datasets (RAVDESS, MEAD, and MELD-FAIR), which primarily focus on English speakers and Western facial expressions. This limits the generalizability of the results and raises concerns about the model's performance in more diverse cultural and linguistic contexts. The lack of evaluation on larger or more varied datasets hinders the paper's ability to demonstrate the framework's robustness and applicability to real-world scenarios.

**Questions For Authors:**

Please see the weakness.

**Relation To Broader Scientific Literature:**

None

**Theoretical Claims:**

It seems the theoretical claims are correct in this paper.

---

> ### Author Rebuttal · Authors · 2025-04-01
>
> Thank you very much for your positive comments and efforts in reviewing our manuscript. We try our best to address your questions as follows.
>
> **Q1: Limited novelty**
>
> Thank you for the opportunity to clarify the distinctions from previous methods. Although DEmoFace builds on existing discrete diffusion models (DDMs), our work makes key contributions.
>
> 1. While DDMs have shown great promise in text generation, their potential for speech generation—particularly with RVQ codec tokens—remains underexplored.
> 2. Existing DDMs struggle with effective guidance for multi-conditional generation. Our EPFG addresses this challenge by providing more efficient guidance, supported by strong empirical evidence, while also introducing theoretical insights that enhance the understanding of multi-conditional generation in DDMs.
> 3. DEmoFace is a unified framework for both acoustic- and visual-guided speech generation, with extensive results demonstrating its efficiency.
>
> **Q2: Limited data size**
>
> To address concerns about model generalization, we randomly sample 1,500 utterances from the LRS2 benchmark to evaluate its performance on real-world out-of-domain data.  Results demonstrate that DEmoFace consistently achieves high-quality generation and outperforms the Face-TTS, even for unseen speakers and content.
>
> | Methods      | EmoSim$\uparrow$ | SpkSim$\uparrow$ | RMSE$\downarrow$ | MCD$\downarrow$ | WER$\downarrow$ |
> | ------------ | ---------------- | ---------------- | ---------------- | --------------- | --------------- |
> | Face-TTS     | 0.64             | 0.13             | 104.39           | 14.29           | **16.60**       |
> | **DEmoFace** | **0.75**         | **0.64**         | **96.50**        | **12.75**       | 20.26           |
>
> Furthermore, as suggested by Reviewer 7YBQ, we have conducted dataset-wise experiments on LRS2, performing full fine-tuning of the base model for 20 epochs on four datasets to capture dataset-specific knowledge. The results are as follows:
>
> | Methods          | $\text{Num}_\text{utterance}$ | $\text{Num}_\text{word}$ | $\text{Num}_\text{speaker}$ | EmoSim$\uparrow$ | SpkSim$\uparrow$ | RMSE$\downarrow$ | MCD$\downarrow$ | WER$\downarrow$ |
> | ---------------- | ----------------------------- | ------------------------ | --------------------------- | ---------------- | ---------------- | ---------------- | --------------- | --------------- |
> | Finetune-RAVDESS | 1,140                         | 25                       | 19                          | 0.57             | 0.56             | 112.68           | 13.74           | 92.05           |
> | Finetune-MELD    | 2,150                         | 2,996                    | 143                         | 0.72             | 0.53             | 105.82           | 13.08           | 38.16           |
> | Finetune-MEAD    | 8,876                         | 6,504                    | 36                          | 0.70             | 0.50             | 102.47           | 13.06           | 33.64           |
> | Finetune-LRS3    | 14,601                        | 15,401                   | 719                         | 0.72             | 0.63             | 101.58           | 12.86           | 20.73           |
> | **DEmoFace**     | 26,767                        | 15,545                   | 917                         | **0.75**         | **0.64**         | **96.50**        | **12.75**       | **20.26**       |
>
> We have four key findings: 1) limited semantic content (low utterance and word counts) leads to higher WER; 2) limited speaker diversity (a small number of speakers) negatively affects SpkSim; 3) emotion-rich but small datasets like RAVDESS may not accurately reflect real-world distributions, as acted emotions tend to be exaggerated; and 4) mixed-dataset training improves generalization across all aspects on out-of-domain real-world data.
>
> In future, we will expand the dataset to enhance diversity and real-world applicability.
>
> **Q3: Clarification on efficiency analysis**
>
> We measure latency on 4090 GPU with a mini-batch size of 1 and 32 utterances, and function evaluation numbers (NFE) of 32 and 64. Latency is averaged over the test set utterances, and we report the Real-Time Factor (RTF), which indicates the time (in seconds) required to synthesize one second waveform. The results show that our method with NFE=32 and batch size=1 has the potential to build a real-time TTS system, being 2 times faster than real-time. In future, we will optimize inference efficiency by developing accelerated sampling techniques.
>
> |                                  | RTF$\downarrow$ |
> | -------------------------------- | --------------- |
> | VoiceCraft (batch size=1, NFE=1) | 1.92            |
> | ChatTTS (batch size=1, NFE=1)    | 0.30            |
> | Ours (batch size=1, NFE=32)      | 0.49            |
> | Ours (batch size=1, NFE=64)      | 0.96            |
> | Ours (batch size=32, NFE=32)     | 0.13            |
> | Ours (batch size=32, NFE=64)     | 0.23            |

---

### Official Review · Reviewer_mfqr · 2025-03-18

**Overall Recommendation:** 3

**Summary:**

The paper introduces Emotional Face-to-Speech (eF2S), a novel task that synthesizes emotional speech solely from expressive facial cues. The authors propose DEmoFace, a generative framework leveraging a discrete diffusion transformer (DiT) with curriculum learning, integrated with a multi-level neural audio codec. Key contributions include a multimodal DiT block, a coarse-to-fine curriculum learning strategy, and an enhanced predictor-free guidance mechanism. Experimental results demonstrate improved naturalness and emotional consistency compared to baselines, even surpassing speech-driven methods.

## update after rebuttal

Thanks for the clarification. I maintain my score.

**Claims And Evidence:**

The claim that eF2S generates emotional speech purely from facial expressions is well-supported by experiments.
The introduction of a discrete diffusion model and curriculum learning is novel and backed by strong empirical evidence.
However, the claim that DEmoFace surpasses speech-driven models needs further justification—some results (e.g., WER) still favor speech-guided methods.

**Essential References Not Discussed:**

The references are sufficient.

**Experimental Designs Or Analyses:**

The dataset selection is reasonable, covering RAVDESS, MEAD, MELD-FAIR, and a subset of LRS3.
The comparison with state-of-the-art models (e.g., Face-TTS, EmoSpeech) is thorough.

**Methods And Evaluation Criteria:**

The proposed method is well-structured, with clear pipeline descriptions.
The use of curriculum learning to gradually introduce high-level tokens is innovative.
The evaluation metrics (WER, MCD, EmoSim, SpkSim) are appropriate.

**Other Comments Or Suggestions:**

Add ablation studies on dataset quality and robustness.

**Other Strengths And Weaknesses:**

Strengths:

•	Novel problem formulation (eF2S).

•	Strong empirical results with extensive benchmarks.

•	Open-source potential for future research.

**Questions For Authors:**

1. How does the model handle ambiguous facial expressions (e.g., neutral vs. mild happiness)?

2. How would DEmoFace perform on out-of-domain data (e.g., unseen speakers, languages)?

3. What are the training costs and computational efficiency compared to standard TTS models?

**Relation To Broader Scientific Literature:**

The paper builds upon previous face-driven TTS, emotional TTS, and discrete diffusion models.

**Theoretical Claims:**

The paper extends diffusion models to multimodal emotional speech generation, which is a promising direction. The enhanced predictor-free guidance (EPFG) mechanism is theoretically well-motivated, but the justification for its superiority over standard PFG needs further clarity.

---

> ### Author Rebuttal · Authors · 2025-04-01
>
> We are grateful for your positive feedback and constructive suggestions, and try our best to address your concerns as follows.
>
>
>
> **Q1: Facial expression ambiguity**
>
> Thank you for your insightful comment. In this paper, we leverage a pre-trained facial expression recognition model to generate one-hot emotion labels for learning emotional embeddings, serving as a plug-and-play component. This allows for the integration of stronger models, such as micro-expression recognition, to enhance expression precision and reduce expression ambiguity. We will incorporate this discussion in the future version.
>
>
>
> **Q2: Limited data size and multilingual extension**
>
> Thank you for your insightful comment. To address concerns about model generalization, our submitted paper ensures that test set speakers are unseen for RAVEDSS and MEAD, demonstrating strong generalizability. In addition, per Reviewer kkEx's suggestion, we have supplemented an out-of-domain evaluation on LRS2 by randomly sampling 1,500 utterances, with results provided in the Q2 response to Reviewer kkEx. Experimental results show that DEmoFace achieves better generalizability and outperforms the Face-TTS even with unseen speakers and semantic content.
>
> Expanding to unseen languages is an interesting challenge that requires multilingual datasets and handling linguistic and phonetic differences. However, this is beyond the scope of our eF2S task, which focuses on generating speech aligned with facial identity and emotional expression. In the future, we plan to explore multilingual speech generation using International Phonetic Alphabet (IPA) embeddings.
>
>
>
> **Q3: Training computational efficiency with standard TTS model**
>
> Thank you for your valuable suggestion. Following DiTAR [1], which evaluates the FLOPs of standard Non-AutoRegressive (NAR) TTS models, we compare DEmoFace with both Continuous (Cont.) and Discrete (Disc.) NAR methods at Number of Function Evaluations (NFE = 32). As shown in the following table, DEmoFace achieves comparable training efficiency to other NAR TTS models. In the future, we will further optimize its efficiency and extend DEmoFace from cross-modal face-to-speech to standard TTS to explore its broader impact.
>
> | Type      | Methods                 | Params | TFLOPS |
> | --------- | ----------------------- | ------ | ------ |
> | Cont. NAR | E2-TTS (NFE=32)         | 0.3B   | ~56.5  |
> | Cont. NAR | F5-TTS (NFE=32)         | 0.3B   | ~37.4  |
> | Disc. NAR | MaskGCT (NFE=50)        | 1.1B   | ~116.7 |
> | Disc. NAR | NaturalSpeech 3 (NFE=1) | 0.5B   | ~8.9   |
> | Disc. NAR | **DEmoFace** (NFE=32)   | 0.2B   | ~12.9  |
>
> [1] DiTAR: Diffusion Transformer Autoregressive Modeling for Speech Generation
>
>
>
> **Q4: Comparison with acoustic-driven methods**
>
> In Table 1 of the main paper, the key difference between acoustic-guided and visual-guided methods lies in the modality used to model timbre and emotional prosody, as reflected in SpkSim and EmoSim. However, both approaches rely on textual modality to guide semantic content generation, as indicated by WER. Our acoustic-guided DEmoFace\* outperforms the visual-guided DEmoFace due to a smaller distribution shift from ground-truth (GT) speech. This is because DEmoFace\* is trained and tested with speech modality, while DEmoFace is trained with GT speech but tested with vision modality.
>
>
>
> **Q5: Superiority of EPFG**
>
> The key difference between EPFG and vanilla PFG in discrete diffusion lies in multi-condition disentanglement versus aggregation. As stated in the main text (lines 401 and 432), EPFG significantly enhances multi-conditional generation quality, mitigating semantic confusion caused by aggregation. We will clarify this more explicitly in the future version.

---

### Decision · Program_Chairs · 2025-05-01

**Decision:**

Accept (poster)

**Comment:**

The paper introduces Emotional Face-to-Speech (eF2S) as a novel task that synthesizes emotional speech solely from expressive facial cues. Then, the DEmoFace, a generative framework leveraging a discrete diffusion transformer (DiT) with curriculum learning,  is proposed, which is integrated with a multi-level neural audio codec. Key contributions include a multimodal DiT block, a coarse-to-fine curriculum learning strategy, and an enhanced predictor-free guidance mechanism. Experimental results demonstrate improved naturalness and emotional consistency compared to baselines, even surpassing speech-driven methods. The paper can be further improved by adding experiments to validate the generalizability and do ablation studies to justify the importance of facial emotion in speech generation.